

# Dissolved inorganic nitrogen in a tropical estuary at Malaysia: transport and transformation

Shan Jiang[1], Moritz Müller[2], Jie Jin[1], Ying Wu[1], Kun Zhu[1], Guosen Zhang[1], Aazani Mujahid[3], Tim Rixen[4], Mohd Fakharuddin Muhamad[3], Edwin Sien Aun Sia[2], Faddrine Holt Ajon Jang[2], Jing Zhang[1]

5 [1] State Key Laboratory of Estuaries and Coastal Research, East China Normal University, 200062, Shanghai, China
[2] Faculty of Engineering, Computing and Science Swinburne, University of Technology, Sarawak Campus, Malaysia
[3] Faculty of Resource Science & Technology, Universiti Malaysia Sarawak, 94300 Kota Samarahan, Sarawak, Malaysia
[4] Leibniz Centre for Tropical Marine Research, Fahrenheitstr. 6, 28359 Bremen, Germany

*Correspondence to*: Shan Jiang (sjiang@sklec.ecnu.edu.cn)

10  **Abstract.** Dissolved inorganic nitrogen (DIN), including nitrate, nitrite and ammonium, frequently acts as the limitation for primary productivity. Our study focused on the transport and transformation of dissolved inorganic nitrogen in a tropical estuary, i.e. Rajang river estuary, in Borneo, Malaysia. Three cruises were conducted in August 2016, February-March and September 2017, covering both dry and wet seasons. Before entering the coastal delta, decomposition of the terrestrial organic matter and the subsequent soil leaching was assumed to be the main source of DIN in the river water. In the estuary, 15  decomposition of dissolved organic nitrogen was an additional DIN source, which markedly increased DIN concentrations in August 2016 (dry season). In the wet season (February 2017), ammonium concentration showed a relatively conservative distribution during the mixing and nitrate addition was weak. In September 2017 (dry season), La Niña induced high precipitation and discharge rates, decreased reaction intensities of ammonification and nitrification and hence the distribution of DIN species in the estuary water was similar with the trend found in the wet season. The magnitude of riverine DIN flux 20  varied between 77.2 and 101.5 ton N d$^{-1}$, which might be an important support for the coastal primary productivity.

## 1 Introduction

Nitrogen (N) is an essential element for life. The concentration of N may significantly influence species composition and diversity in terrestrial, freshwater and ocean ecosystems (Vitousek et al., 1997). Apart from nitrogen gas ($N_2$), N is bioactive with highly variable chemical forms. Dissolved inorganic nitrogen (DIN), including nitrate ($NO_3^-$), ammonium ($NH_4^+$) and 25  nitrite ($NO_2^-$), can be easily assimilated by terrestrial plant, algae and bacterial communities (Seitzinger et al., 2002). In addition, due to the high solubility, DIN can be easily transported among ecosystems as a part of the hydrologic cycle (Galloway et al., 2004). Consequently, DIN is assumed to be the most active component in the N cycle and the transport of DIN among ecosystems is a hotspot in biogeochemical research on the global scale (Gruber and Galloway 2008).

Estuaries are the linkage between terrestrial surface water and coastal sea (Seitzinger et al., 2002) and received great 30  attention from researchers and coastal managers with regard to the quantification of terrestrial DIN transport (e.g. Falco et al.,



2010; Holmes et al., 2012; Li et al., 2013; Kuo et al., 2017). The mixing between fresh and saline water develops substantial physiochemical gradients. They can be seen in various parameters such as dissolved oxygen (DO), salinity and pH (Spiteri et al., 2008), which influence the growth of distinct bacterial communities (e.g. Goñi-Urriza et al., 2007; Spietz et al., 2015). These bacteria actively participate in DIN transformation processes, such as denitrification, DNRA (dissimilatory nitrate

reduction to ammonium) and Anammox (anaerobic ammonium oxidation; Burgin and Hamilton, 2007). These processes strongly influence DIN concentrations (Canfield et al., 2010), adding uncertainties to the precise estimation of DIN fluxes. Moreover, environmental parameters related with these gradients vary significantly between seasons, leading to a highly dynamic DIN export. Additionally, riparian regions are the focus of intensive anthropogenic processes, such as agriculture, manufacture and wastewater treatment (Richardson et al., 2007). These activities strongly influence DIN cycling in rivers

and their estuaries.

Researchers increased the sampling frequency and introduced regional modeling work to improve the understanding of DIN transport and transformation processes. More importantly, stable isotope ratios, e.g. $\delta^{15}$N-NO$_3^-$, $\delta^{18}$O-NO$_3^-$, $^{15}$N-PN (particle nitrogen), have been introduced to trace DIN transport and transformation processes (Middelburg and Nieuwenhuize, 2001). Stable isotope technique has been applied in a number of estuaries located in temperate and sub-tropical zones, such as

Changjiang estuary (China; Yu et al., 2015; Yan et al., 2007), Seine River estuary (France; Sebilo et al., 2006), River Thames estuary (U.K.; Middelburg and Nieuwenhuize, 2001) and Werribee River estuary (Australia; Wong et al., 2014). The delineated reaction pathways and DIN sources and sinks from these research outcomes largely improved our understanding of DIN cycle, which is crucial for projections as well as the regulation/law enactment.

Despite the significant advances made, knowledge gaps in DIN transport via estuaries still exist with geographic coverage as

one of the major gaps. As aforementioned, previous research work was intensively conducted in temperate and sub-tropical zones. Tropical zone, characterized by the high annual temperature and intensive precipitation, hosts substantial rivers and streams; while the studies on DIN transport are scarce. The lack of information potentially hampers the holistic understanding of DIN transport from rivers to the ocean and increases uncertainties in the global DIN budget estimation (Voss et al., 2013). In addition, since the Second World War, a raid development in tropical countries has been witnessed,

suggesting that the tropical zone is becoming a hotspot for DIN production, utilization and transport (Seitzinger et al., 2002). The ecological and environmental response in tropical estuaries on the DIN related anthropogenic pressure is less documented. Moreover, the coastal ecosystem in the tropical zone is often complex and harbors substantial seagrass meadow, fish species and coral reefs (Sale et al., 2014). The enhanced intrusion of allochthonous DIN from estuaries to coastal region might be an ecological risk for local systems (Barbier et al., 2011). Adding these together, it is of urgency and importance to

apply robust DIN studies in the tropical zone and cooperation between researchers from multiple disciplines is highly necessary.

In the present study, three cruises in a typical tropical estuary, i.e. Rajang river estuary (hereafter Rajang estuary), in Malaysia were conducted, from August 2016 to September 2017, covering both dry and wet seasons. Each cruise started from upper stream stations and extended to the coastal ocean. Concentrations of PN, dissolved organic nitrogen (DON) and



DIN were determined, and isotope fractions of $^{15}$N-NO$_3^-$ and δ$^{18}$O-NO$_3^-$ as well as $^{15}$N-PN were analyzed accordingly. The main research aims for the present study included (1) identification of DIN sources in the river water; (2) exploration of transfers and reactions with regard to DIN at different tributaries in the Rajang estuary and influences on DIN concentration; (3) estimation of magnitude of DIN fluxes injected from Rajang to coastal ocean.

## 2 Materials and methods

### 2.1 Study site

Rajang river (hereafter Rajang) is located in Sarawak state, Borneo (Fig.1 A and B). Sarawak is one of the largest states in Malaysia with an intensive tropical forest coverage. By 2000, the population in Sarawak was 2.5 million with the urbanization level of 47.9% (https://www.sarawak.gov.my/web/home/article_view/240/175). The climate in the Sarawak state is tropical ever-wet (Staub et al., 2000) and frequently influenced by El Niño-Southern Oscillation (ENSO) and Madden-Julian Oscillation (Sa'adi et al., 2017). The annual precipitation in the Rajang watershed is approximately 4000 mm, especially in the period from November to next February due to Indian Ocean Monsoon (Müller et al., 2015). As a result, this period is usually identified as the wet season and the remaining months are attributed to the dry season. The temperature variation was limited between each month and the highest mean daily temperature reached 33 ℃ (Ling et al., 2017).

Rajang is the longest river in Malaysia with a length of ca. 530 km. It originates from the Iran Mountains, flows through several cities, such as Kanowit, Song and Sibu, enters Rajang Delta and discharges into the South China Sea (Staub et al., 2000). The watershed coverage is approximately $5.1 \times 10^4$ km$^2$. The river bed in the upper stream was mainly composed by Creaceous-Eocene age sediments. Igneous intrusive and extrusive rocks were observed along the river (Staub et al., 2000). Coupled with precipitation and erosion, suspended particles in the Rajang water were frequently over a level of 200 mg L$^{-1}$ (Ling et al., 2017). After accumulation for 6000 years, the sediment particles from the upper stream developed a large area of alluvial delta (from Sibu to estuary mouths). The delta plain is mainly composed by siliciclastic sediments (bottom) and organic matter enriched sediments (surface). A substantial fraction of surface sediments was identified as peat deposits with a maximum depth of 15 m (Staub et al., 2000). There are four major tributaries in the Rajang Delta, namely Rajang, Hulu Seredeng (further separates into two tributaries: Paloh and Lassa), Belawai and Igan (Fig. 1 B). Apart from Rajang tributaries, the peat deposits were frequently observed in the remaining three tributaries (Fig. 1 C). The tidal range in the Rajang estuary is from 2.8 to 5.6 meter and decreases from Rajang (macro tidal range) to Igan (meso tidal range). The saltwater intrusion could reach the downstream of Sibu with a few kilometers distance to the city (Müller-Dum et al., 2018). The total discharge rate of Rajang was estimated to be 3000 to 6000 m$^3$ s$^{-1}$, depending on season. The river water mainly injected through the Igan branch to the South China Sea (Jakhrani et al., 2013). Rajang and its delta play an important role in the national economy. Fishery, logging and timber processing are the traditional supports for local citizens (Salam and Gopinath, 2006; Miettinen et al., 2016). In addition, industry plantation for oil palm and acacia boomed in recent years (Lam

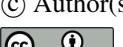


et al., 2009), occupying more than 50% peatland (11% of the catchment size) in Rajang watershed (Miettinen et al., 2016). These activities have already led to patchy deforestation, from upper steam to the coastal delta (Fig. 1 C).

## 2.2 Cruises and sample collection

Three cruises were conducted during 2016-2017, covering two dry seasons (August 2016, September 2017) and a wet season
(February to March 2017). The sampling in each cruise included the river water sites, and brackish water sites in Igan, Lassa/Paloh and Rajang tributaries. High salinity water samples (>30‰) from the adjacent coastal ocean were also collected during each survey. The total sampling sites in each cruise ranged between 16 and 32 stations. In September 2017, additional pore water samples from the edge of mangrove forest, peatland and coastal sandy beach were collected. Rainwater was gathered once during the September cruise.

The river water and coastal seawater were collected into 1 L acid-prewashed HDPE sampling bottles via a pole-sampler that decreases the contamination from boat surface and engine cooling water (Zhang et al., 2015). Apart from four stations in September 2017, salinity, temperature, DO and pH were measured *in-situ* by an Aquaread® multiple parameters probe (AP-2000). For pore water samples, a sampling hole with a depth of approximately 20 cm was dug during a low tide. The seeped pore water that accumulated in the bottom of the sampling hole was discarded three times before collection. Subsequently,
pore water was sucked into a 50 mL syringe and then transferred to 250 mL acid-prewashed sampling bottles. The rainwater was collected under the roof at a local primary school in a strong precipitation event. The rainwater from the first 10 min was discarded. Salinity of the pore water and rainwater was determined by a refractometer.

The filtration was conducted immediately after sampling. The harvest water samples were shaken and then divided into two portions (excluding pore water and rainwater). The first portion was filtered via polycarbonate membrane filters (0.4 μm
pore size, Whatman®) into 60 mL sampling bottles for the determination of dissolved nitrogen species in three cruises and $NO_3^-$ isotope fractions ($^{15}N$-$NO_3^-$ and $^{18}O$-$NO_3^-$) in the last two cruises. The other portion was filtered via pre-combusted (450 ℃, 4 hours) glass fiber filters (average pore size 0.7 μm, Whatman®) for the analyses of SPM and PN concentration, as well as $\delta^{15}N$-PN. Both liquid and membrane samples were kept at -20 ℃ environment until laboratory analyses.

## 2.3 Mixing experiments

In September 2017, a mixing experiment was conducted to explore the influence of river-borne suspended particles on DIN transformation along the salinity gradient. In particular, river water samples (salinity: 0) from Sibu (10 km downstream from the city dock) and coastal ocean (salinity: ca. 32‰) were collected. All the seawater and half of the river water were filtered through polycarbonate membrane filters for the removal of particle matters. The first treatment group was assigned to be the mixture between the filtered seawater and the particle-free river water. In practice, they were mixed and placed in 1 L acid
prewashed HDPE bottles with a total volume of 500 mL. The percentage of river water in the system was 0% (purely filtered seawater), 25%, 50%, 75% and 100% (pure river water). The second treatment group contained filtered seawater and unfiltered river water, while the total volume and percentage of river water were identical with the first treatment. The HDPE




bottles were placed in the darkness at 25-26 °C for 24 hours. During the incubation, all the bottles were rotated to sustain the suspension of particles. Afterwards, all the mixture was filtered again and the liquid was stored in 60 mL bottles in a freezer for the determination of DIN, DON, $\delta^{15}N\text{-}NO_3^-$ and $\delta^{18}O\text{-}NO_3^-$.

## 2.4 Laboratory analyses

After thawing and thoroughly remixing, concentrations of $NH_4^+$, $NO_2^-$ and $NO_3^-$ were determined on a flow-through injection system (SKALAR Analytical B.V., The Netherlands) using standard colorimetric methods (Grassholf et al., 2007) after modification by manufactures. The determination limit for these species was below 0.1 µM and analysis pression was ca. 3.5%. The content of total dissolved nitrogen (TDN) was measured by the potassium persulfate digestion method (121 °C, 30 min digestion) according to Ebina et al. (1983). The difference in concentration between TDN and DIN was assumed to be

the level of dissolved organic nitrogen (DON). Both $\delta^{15}N\text{-}NO_3^-$ and $\delta^{18}O\text{-}NO_3^-$ were determined using the bacterial reduction method on the basis of Weigand et al. (2016). In practice, $NO_3^-$ in the water samples were transformed into $N_2O$ via the denitrifier (*P. aureofaciens*; Sigman et al., 2001) after removal of $NO_2^-$ by sulfanilamide acid solution (Weigand et al., 2016). The produced $N_2O$ was injected into Thermo-Fisher Precon system (Thermo Fisher, USA) and then flowed through Finngan chromatographically loops, finally into a Thermo-Fisher Delta V isotope system. The calculation from $N_2O$ isotope fraction

to $\delta^{15}N\text{-}NO_3^-$ and $\delta^{18}O\text{-}NO_3^-$ followed Casciotti and Mcilvin (2007). The determination limit for the original $NO_3^-$ concentration was approximately 0.9 µM. The method precession was ca. 0.2‰ ($\delta^{15}N\text{-}NO_3^-$) and 0.5‰ ($\delta^{18}O\text{-}NO_3^-$), respectively.

## 2.5 Mathematical analyses

To understand the addition or removal of solutes during the mixing, a two-endmember mixing model (Liss, 1976) used for
the conservative distribution of solute concentration and related isotope fractionations was invoked:

$$N_{mix} = f_r \times N_r + (1 - f_r) \times N_o, \tag{1}$$

where $N_{mix}$ is the conservative concentration of a specific N solute at a particular salinity; $N_r$ is the solute concentration at the fresh water endmember (obtained from the starting site in each tributary); $N_o$ is the concentration at the ocean endmember (the high salinity site near each tributary outlet); $f_r$ is the fraction of fresh river water, which is calculated as:

$$f_r = (S_o - S_{mix})/(S_o - S_r), \tag{2}$$

where $S_{mix}$ is the sample salinity; $S_r$ and $S_o$ are salinity at the river and saline water, respectively. For the conservative isotope fraction in each sample, it can be calculated as:

$$\delta_{mix} = [f_r \times N_r \times \delta_r + (1 - f_r) \times N_o \times \delta_o]/N_{mix}, \tag{3}$$

where $\delta_{mix}$ is the conservative isotope fraction, such as $\delta^{15}N\text{-}NO_3^-$ or $\delta^{18}O\text{-}NO_3^-$; $\delta_r$ and $\delta_o$ are isotope values at the river
and seawater, respectively. The difference in concentration between the conservative and the observed level was defined as the offset concentration, i.e. an indicator for benthic N reactions. Considering the limited distance and similar environmental





settings between Polah and Lassa tributary, the measurement data from these two tributaries were merged together in the calculation and related plots.

All the statistical analyses, such as student t-test and linear regression were conducted in Minitab 17.0 (Minitab Inc., Pennsylvania State University, U.S.). The significant threshold for all the analyses was α=0.05.

## 2.6 Data plot

The spatial distribution of each parameter, e.g. salinity, temperature and solute concentration, was plotted in Surfer 14.0 (Golden Software Inc., USA) and the dot plots were done in Sigmaplot 12.5 (Systat Software Inc., USA). Given the limited space, a portion of plots was displayed in supplementary materials.

## 3 Results

### 3.1 Water chemistry

In August 2016, the salinity in the sampled water ranged from 0.02 to 31.2‰ (Fig. 2, left panel). Similar salinity range was observed in the remaining cruises. The water temperature ranged from 27.7 to 31.8 °C in August 2016 and the variation among cruises was also limited. The pH varied from 6.03 to 8.12 and displayed a steady increase from river water to coastal ocean water (Fig. S1 and S2). DO fell in the range between 8.2 and 2.7 mg $L^{-1}$. The DO content in the majority of sites was unsaturated, especially in the August cruise. Accordingly, the apparent oxygen utilization (AOU), i.e. the difference between DO saturation level and observed DO concentration, was calculated. It varied from 0.6 and 132 μmol $L^{-1}$ and decreased from fresh river water to coastal saline water (Fig. S3). In the light of the seasonality, the mean AOU at fresh river water and channel water in three tributaries in August 2016 was significantly higher than the values from the remaining two cruises.

### 3.2 Particle nitrogen and its isotope fraction

In August 2016, the SPM concentration ranged from 24 to 120 mg $L^{-1}$ (Fig. 2). Compared with the coastal ocean, the SPM content in river water was markedly higher; while the peak was found in the mixing zone. In the cruises at 2017, SPM concentration in river water markedly elevated, exceeding a level of 500 mg $L^{-1}$. By contrast, the SPM content in the ocean endmember was similar among cruises. The percentage of PN in SPM varied between 0.1% and 0.6% and showed a weak correlation with salinity (Fig. S4). For the PN concentration, i.e. the multiplication product of SPM concentration and PN percentage, varied from 0.07 to 0.57 mg $L^{-1}$. Due to the low concentration of SPM, the range of PN concentration in August 2016 was lower than that obtained from February 2017 and September 2017. The $\delta^{15}$N-PN from three cruises was from 1.9‰ to 11.8‰ (Fig. 2). Apart from several sites in the mixing zone, $\delta^{15}$N-PN in fresh river water and coastal ocean water was similar. By contrast, a clear seasonal variation in $\delta^{15}$N-PN was found. The highest value was observed in September 2017, while the lowest fraction was obtained in February 2017 (Fig. 2).





### 3.3 Dissolved nitrogen and related isotope fractions

Compared with PN concentration, the content of dissolved fractions was relatively minor. In August 2016, the DON concentration varied from 2.6 to 14.8 µM and high levels were found in estuary channels (Fig. S6). In the remaining two cruises, excluding several sites with high levels that were patchily distributed in tributaries in September 2017, a similar

concentration range was obtained. The $NH_4^+$ concentration was 0.39 to 17.3 µM (Fig. 3). Similar with DON, the seasonal variation in $NH_4^+$ concentration was limited. During the mixing, a slight increase in $NH_4^+$ concentration in August cruise was found (Fig. 4), especially in the Lassa tributary; while the remaining two cruises showed a limited variation during the river-ocean mixing. For $NO_3^-$ concentration in the river water, it varied between 1.6 and 14.8 µM in August 2016 (Fig. 3). The concentration in the remaining two cruises slightly decreased. The highest concentration was below the level of 10 µM. The

similarity derived from $NO_3^-$ concentration distribution in 2017 cruises led to an insignificant variation in the range for $\delta^{15}N$-$NO_3^-$ and $\delta^{18}O$-$NO_3^-$ between seasons (Fig. 3). In terms of $NO_2^-$, i.e. the minimum component in DIN inventory, the concentration varied between 0.09 and 3.3 µM in August 2016. Similar levels were found in the remaining samples.

Along the salinity gradient, positive offset for $NO_3^-$ concentration was frequently observed, especially in Lassa and Igan branches, which suggests net generations during the mixing (Fig. 5). Coupled with $NO_3^-$ generation, negative offsets in $\delta^{15}N$-

$NO_3^-$ and $\delta^{18}O$-$NO_3^-$ at most sites were observed (Fig. 5). The relationship between offsets of $\delta^{15}N$-$NO_3^-$ and $\delta^{18}O$-$NO_3^-$ were linearly correlated (Fig. 5). The slope was 0.99 and $R^2$ was 0.63 ($p<0.05$).

### 3.4 Mixing experiments

In the particle-free (filtered) group, the mean DON concentration was 7.1 µM in river water and 4.7 µM in seawater (Fig. 6). During the mixing, DON concentration slightly departed from the conservative distribution. Similar trend was found in

unfiltered (particle-contained) group. Different from DON, $NH_4^+$ concentration in both groups were nearly conservative and the difference in concentration between groups was minor. For $NO_3^-$ content, apart from the seawater (identical between groups), the concentration in the particle-contained group was markedly lower than that in the filtered group. As a mirror of concentration variation, $\delta^{15}N$-$NO_3^-$ and $\delta^{18}O$-$NO_3^-$ was elevated in the particle-contained group. Furthermore, compared to the conservative distribution, a deficit in $NO_3^-$ content in brackish water (salinity: 24‰) at the unfiltered group was obtained.

Concurrently, an increase in both $\delta^{15}N$-$NO_3^-$ and $\delta^{18}O$-$NO_3^-$ at the identical salinity was observed. In terms of $NO_2^-$, the unfiltered group showed a strong removal during the mixing while the filtered group maintained a conservative distribution (Fig. 6).

### 3.5 Pore water and rainwater

Salinity of the collected rainwater was 0 (Table 1). The concentration of $NH_4^+$ was 18.9 µM. The $NO_3^-$ level was 16.4 µM

with a markedly high level of $\delta^{18}O$-$NO_3^-$ (55.3‰) compared to the river water. For the pore water samples, the salinity varied from 1.0 to 21.5‰. These samples were enriched with $NH_4^+$ (22.8 to 121 µM) and DON (34.5 to 89.2 µM). In





comparison, the level of $NO_3^-$ and $NO_2^-$ in pore water samples was limited. The isotope fraction was similar with the river water.

## 4 Discussion

### 4.1 DIN sources in fresh river water

Rajang is an aorta in Sarawak and receives substantial materials from its watershed. In the river water (salinity: 0), the proportion of DIN in riverine N inventory was minor, accounting for ca. 20% to 30% in total N inventory (Fig. 7A). In comparison to rivers located in dense population regions, such as Pearl River in China, Mississippi River in the U.S.A., Danshui River in Taiwan, China, and the Mekong River in Vietnam, concentrations of $NO_3^-$ and $NH_4^+$ in Rajang water were low (Table 2).

As shown in Fig. 7B, the signal of $\delta^{18}O\text{-}NO_3^-$ and $\delta^{15}N\text{-}NO_3^-$ highlights that the decomposition of the terrestrial organic matter and its subsequent leaching from soils was an important source of $NO_3^-$ in river waters. Despite the relatively low DIN concentration, the DIN yield of the Rajang was higher than those of other tropical rivers due to higher ENSO induced rain fall and the resulting high surface water discharges (Fig. 7C, D). Among the tropical rivers around the South China Sea, e.g. Mae Klong river in Thailand and Langat river in Malaysia (Malaysian Peninsular), the yield in $NH_4^+$ and $NO_3^-$ in Rajang

at August 2016 was higher (Fig. 7C). The $NH_4^+$ production is even higher than the Wanquan river (China), which features intensive human activities and a large number of tourists. Such high yield may result from the abundant storage of organic matter and ammonification induced $NH_4^+$ accumulation in Sarawak peatland (Melling et al., 2007). Moreover, DIN concentrations in Rajang exceeded those detected in pristine tropical black water rivers (Baum and Rixen, 2014), suggesting a potential influence from anthropogenic activities. In Sarawak, the chemical fertilizer requirement in oil palm plantation

(Tarmizi and Mohd Tayeb, 2006) may be the most likely DIN source since the unconsumed fertilizer likely drained into Rajang water coupled with precipitation.

Notably, the level of $NO_2^-$ and $NO_3^-$ in fresh river water was stable among cruises, while $NH_4^+$ concentration significantly varied in two dry seasons. $NH_4^+$ can be the reaction output of DON and PN mineralization and subsequent ammonification. The transformation can be performed in the terrestrial aquifer and/or Rajang. Among cruises, the concentration of DON was

stable. The level of PN increased in September 2017 (Fig. 7A). Consequently, the production of $NH_4^+$ may not be constrained in Rajang water and the drop of $NH_4^+$ concentration in Rajang resulted from the declined $NH_4^+$ production in terrestrial soils.

A strong El Niño event was observed from January to June 2016, the Niño 3.4 Index reached 2.5 (threshold 0.5). Subsequently, La Niña occurred and introduced stronger precipitation in Malaysia (Fig. 7D). Consequently, the weather in

September 2017 was comparably 'wet' than the dry season in August 2016. Abera et al. (2012) revealed a significant reduction in extractable $NH_4^+$ content in tropical soils when precipitation enhanced. They addressed that the possible reason for the decreases in ammonification intensity was the enhanced moisture in terrestrial soils, because high moisture could



significantly restrain the aeration in peatland (Daniels et al., 2012). A similar phenomenon might also occur in the Rajang watershed, the $NH_4^+$ production in tropical soils likely decreased in September 2017. Concurrently, strong precipitation enhanced river water volume that caused solute dilution. Adding these together, the concentration decline of $NH_4^+$ in Rajang water was found. Such variation reflects the dynamic linkage between global climate events and local N storage.

## 4.2 N transformations in the estuary mixing zone

In August 2016, concentrations of $NO_3^-$, $NO_2^-$ and $NH_4^+$ in the estuary mixing zone were higher than in the Rajang river and the coastal ocean. This concentration increases generally result from (1) direct input from tributary streams or pore water exchange and (2) N transformations. In Rajang Delta, there are several small streams, continuously adding solutes into the estuary water (Staub et al., 2000; Gaslto, 2010). However, the discharge rates from these streams are relatively small compared to the Rajang, and hence their input can be identified as a point source, of minor importance. Precipitation, introducing solutes on a regional scale, could enhance $NH_4^+$ concentration in river water but only within a small temporal scale (Fig. 8A). Alternatively, the exchange between pore water in both cohesive and sandy sediments and surface water can add DIN in river water on a regional scale due to the wide contact (Fig. 8A). In coastal areas, this exchange can be driven by tidal pumping (Santos et al., 2012a). In particular, during high tides, the tidal sediment marsh was flushed by the estuary water. The overlying water seeped into the sediment along the conduits created by crabs and worms, or plant roots. During the ebbing tide, pore water slowly drains and adds solutes (Santos et al., 2012b; Tait et al., 2016). In the current research, $NH_4^+$ concentrations in different pore water samples were significantly higher than the level found in the estuary water. Coupled with the macro-meso tides in Rajang estuary, the magnitude of $NH_4^+$ flux from pore water to the Rajang might be great, as outlined in Fig. 8A. Moreover, the diffusion from the benthic sediment to the overlying water also adds $NH_4^+$ in the estuary water (Fig. 8A). Notably, low $NO_3^-$ levels were found in pore water samples, suggesting a limited effect on the river water $NO_3^-$ concentration.

N transformations, including ammonification, nitrification, DNRA (Burgin and Hamilton, 2007), may also markedly contribute to the enhancement in $NH_4^+/NO_3^-$ concentration. Besides N fixation, ammonification is the only reaction that increases the total DIN concentration, which is coupled with the decomposition of PN and/or DON. In August 2016, the unsaturation of DO and high records of AOU were obtained in the mixing zone, suggesting the occurrence of active aerobic respiration on the basis of organic matter decomposition. Concurrently, the abundant SPM in the estuary water was observed, providing a significant amount of PN (Fig. 2 and Fig. 7A). However, the PN concentration was relatively conservative in the mixing zone and the $\delta^{15}$N-PN variation was limited (Fig. S6). Consequently, PN may not be involved in the biogeochemical reaction and hence it was not the major reactant for the ammonification.

Instead, DON, especially the reactive portion, can serve as an active reactant (Brandes et al., 2007). In the mixing experiment, a reduction in DON concentration was observed, confirming the biogeochemical activity. Despite the injection from sediment pore water in the Rajang estuary (Fig. 8A), the DON concentration in the mixing zone was lower than that in Rajang water and coastal ocean water, indicating a net consumption. DON may continuously transform into $NH_4^+$ via



mineralization and subsequent ammonification (Fig. 8B). Photo-degradation in tropical rivers also accelerates the decomposition of DON and benefits accumulation of $NH_4^+$ in surface river water (Martin et al., 2018). In Lassa branch, a clear DON consumption and the possibility for the occurrence of ammonification was obtained (Fig. 7E and F).

The elevation in the concentrations of $NO_2^-$ and $NO_3^-$ was attributed to nitrification (Fig. 8B). In February 2017, coupled with $NO_3^-$ concentration increases, declines in $\delta^{15}N$-$NO_3^-$ and $\delta^{18}O$-$NO_3^-$ during the mixing were found, which reinforced the occurrence of nitrification (Fig. 9B). In order to describe the intensity of such increasing trend, the reaction factor (f) was introduced for each DIN species on the basis of concentration difference between observed distribution and conservative distribution for each solute (calculation process described in the legend of Fig. S10). The positive value indicates the solute addition during the mixing and vice versa. The magnitude of f is closely linked to the magnitude of reaction intensity in addition/removal at estuary water.

Notably, the enhancement in DIN species varied between tributaries. Compared with Lassa/Paloh and Rajang channels, the distribution of $NO_2^-$ and $NO_3^-$ in Igan tributary during the mixing tended to be conservative and f values were comparatively small (Table S1), while $NH_4^+$ concentration remained to be high during the mixing. In addition, the AOU in the Igan channel was relatively low in comparison to other tributaries. It can be deduced that the pore water exchange process during the mixing still occurred, while the nitrification intensity in Igan tributary was relatively weak, which likely results from the significant difference in hydrologic environments. In particular, the Igan channel was the main freshwater outlet because of the comparatively low tidal amplitude but high discharge rate (Jakhrani et al., 2013). The large freshwater plume pushed the mixing zone towards the coastal ocean (Fig. 2). Consequently, the brackish water could be rapidly diluted by the coastal ocean water, leading to a short residence time for the mixing of brackish water. By contrast, the mixing in other river tributaries occurred in the river channels, causing the slow dilution and long residence time for the brackish water. The difference in water residence time likely created the varied reaction intensity (Zarnetske et al., 2011). In the light of the reactions between Lassa and Rajang tributary, the difference in reaction pattern (addition) and intensity (f value) was small, although the proportion of peatland coverage differed due to deforestation (Fig. 1). In the same estuary, Müller-Dum et al. (2018) also reported a limited difference in $CO_2$ emissions between peat and non-peat area. Such pattern may relate with the temporal scale. In particular, the peatland was the product of 6000 years deposition as aforementioned. The regional deforestation in the estuary occurred in the recent 15 years (Miettinen et al., 2016), indicating that the disturbance has not been developed into the deep region. The influence may not reach an identifiable level. However, the influence from the shrinkage of peatland may enhance in the future.

The distribution of DIN species and related isotopes in the mixing zone between cruises was also observed, indicating the presence of seasonal variability. In February 2017, the traditional wet season in the Sarawak (Müller et al., 2015), the production of $NO_3^-$ and $NO_2^-$ during the mixing was mild compared to August 2016. Accordingly, a decline in magnitude of f was observed, suggesting a weak nitrification intensity in the wet season. Decreases in residence time for the brackish water column due to high river discharge can be the first reason for this decrease. The second factor that influences the generation of $NO_3^-$ and $NO_2^-$ in the estuary water can be SPM related biogeochemical reactions because suspended particles



are versatile. On the one hand, the N content on the particle could release into the water via decomposition (Brandes et al., 2007), subsequently increasing DIN concentration (Fig. 8B). On the other hand, the suspended particles could provide a large number of micro-niches for denitrifiers (Jia et al., 2016), which potentially contribute to the $NO_3^-$ removal during the mixing. Consequently, the addition or removal for $NO_3^-$ content in estuary water likely depends on the reaction capability of
these two controversial pathways.

In Rajang estuary, the PN percentage in SPM frequently ranged from 0.1% to 0.3%, smaller than other tropical rivers located in adjacent regions, e.g. Wonokromo river (0.5%) and Rorong river (0.85%) at Indonesia (Jennerjahn et al., 2004), Godavari river (0.36%; Gupta et al., 1997). In the mixing experiment, small difference in DON concentration between groups, indicating the decomposition of PN was weak. This is in line with the conclusion that PN was inactive in the mixing zone in
previous section. Therefore, the presence of high concentration of PN cannot benefit $NO_3^-$ addition. Alternatively, the denitrification capability evoked by the SPM can be significant due to the large particle surface. In Rajang estuary, suspended particles were enriched in trace metal, such as Fe and Mn (Staub et al., 2000), which potentially accelerates the $NO_3^-$ removal process by serving as electron donor or catalyzer (Burgin and Hamilton, 2007). Furthermore, in estuaries, the presence of flocculation and adsorption attract metal ions on the surface of suspended particles, enhancing the denitrification
potential. In the mixing experiment, the presence of suspended particles markedly decreased levels of $NO_3^-$ and $NO_2^-$ regardless of salinity when compared to the particle-removed group. Additionally, both $\delta^{15}N\text{-}NO_3^-$ and $\delta^{18}O\text{-}NO_3^-$ markedly increased, confirming the presence of denitrification. Moreover, $NH_4^+$ concentration during the mixing was conservatively distributed, indicating the Anammox, that utilizes both $NH_4^+$ and $NO_3^-$, may not be the dominant pathway for $NO_{2+3}^-$ removal.

In September 2017, the end of the dry season according to the historical record, the DIN distribution trend was markedly
different with the pattern from August 2016. Specifically, the generation of $NO_3^-$ and $NO_2^-$ was relatively weak than the pattern in August. For example, the f value was significantly smaller (Table S1) and the $NO_3^-$ concentration offset in three tributaries were markedly lower than these values obtained in August 2016. The variation of both $\delta^{15}N\text{-}NO_3^-$ and $\delta^{18}O\text{-}NO_3^-$ during the mixing process was similar to the trend observed in February 2017. Such pattern likely results from the significantly increased Rajang river discharge in September 2017 due to the continuous occurrence of La Niña events, as
aforementioned. This observation reinforced that the biogeochemical reactions in the tropical zone are mainly constrained by precipitation and the global climate events markedly influences the N transformations on a local scale. Moreover, it is worth noticing that the co-existence of $NO_3^-$ concentration increase and positive $\delta^{15}N\text{-}NO_3^-$ offset in the Rajang branch in the September 2017 (Fig. 5). Apart from the bias introduced from the endmember selection, such distribution indicates the introduction of $NO_3^-$ from human activities. Compared with Lassa and Igan tributaries, Rajang tributary is adjacent to Sirikei
city (Fig. 1C). The anthropogenic activity likely introduced $NO_3^-$ with $^{15}N$ enriched water into Rajang, e.g. wastewater or sewage (Fig. 7B), along with several streams. In the future, coupled with the population increase in Rajang watershed, $NO_3^-$ with anthropogenic signature may increase in Rajang Delta, which should receive more attention.





### 4.3 DIN export fluxes and fate in the coastal ocean

After the consumption/addition in estuaries, DIN injects into coastal oceans. The magnitude of DIN fluxes (Q) that transported to the coastal ocean can be estimated by the following equation:

$$Q = C \times V \times (1 - f) , \tag{4}$$

where $C$ is the mean concentration of DIN species at the fresh river water (Fig. 7), $V$ is the river water discharge (3000 m$^3$ s$^{-1}$ in the dry season, 6000 m$^3$ s$^{-1}$ in the wet season), and $f$ is the mean reaction factor that averaged the reaction factor from the distinct tributary. As outlined in Table S1, the magnitude of $NH_4^+$ fluxes among ranged from 4.57 to 24.7 t N day$^{-1}$ and magnitude of $NO_3^-$ fluxes peaked at 82.4 t N day$^{-1}$. The DIN loading was 77.2 t N day$^{-1}$ in the dry season and 101.5 t N day$^{-1}$ in the wet season.

On a global scale, the DIN delivered from the Rajang to coastal ocean is relatively minor (Table 2). For the rivers around the South China Sea, the magnitude of Rajang-borne DIN fluxes was less than 1/10 of the DIN loading from Mekong River and similar with 1/6 of the DIN from Pearl River. The transported $NO_3^-$ and $NO_2^-$ was rapidly removed in the coastal ocean. Coupled with concentration decrease, $\delta^{15}N$-$NO_3^-$ and $\delta^{18}O$-$NO_3^-$ rapidly increased. In addition, the offset between $\delta^{15}N$-$NO_3^-$ and $\delta^{18}O$-$NO_3^-$ was closely correlated and showed a slope of 0.99 (Fig. 5). Consequently, the decline in $NO_3^-$ concentration

resulted from the consumption of primary productivity (Granger et al., 2010). The consumed $NO_{2+3}^-$ was transferred to organic N, subsequently released into the coastal ocean as $NH_4^+$, due to cell decay and decomposition. The produced $NH_4^+$ may be oxidized to $NO_{2+3}^-$ or utilized by primary productivity again, creating an internal circling (Fig. 8B).

Around Sarawak coastal line, there are abundant tropical coral reefs and fish resources (Hamli et al., 2012; Praveena et al., 2012; Arai, 2015), relying on the phytoplankton and seagrass. The input of DIN likely sustained the growth of the primary

productivity and maintained the ecosystem function. Currently, the riverine DIN input was mild, which did not markedly change the stoichiometry in coastal water. The dominant phytoplankton species was diatom (Saifullah et al., 2014) and harmful algae blooms were not recorded in Sarawak coasts. In addition, the fishery industry is flourished and plays an important role in the local economy (Ikhwanuddin et al., 2011; Hamli et al., 2012). However, the Rajang estuary is subject to increasing human pressures, especially from agricultural fields, fallow shrubland and industrial plantations (Ting and Rose,

2014; Miettinen et al., 2016). Logging and oil palm plantation have resulted in deforestation of peatland and the scale is increasing (Miettinen et al., 2016). The deforestation in Rajang watershed has proved to be a key factor that increases suspended particle concentration and accelerates erosion of terrestrial organic matter (Ling et al., 2017). The long-term influence derived from the human transformations should receive more attention in future studies.

### 5 Conclusions

DIN concentration in the Rajang fresh water was variable between seasons, mainly resulting from the decomposition of terrestrial organic matter. The precipitation might decrease $NH_4^+$ production in the watershed via inhibiting the ammonification intensity. In 2017, La Niña caused increase of precipitation and subsequently lowered the $NH_4^+$ formation





compared to August 2016. This indicates a causal chain between climate and N cycling in tropical soils and rivers. In the estuary mixing zone, pore water exchange and decomposition of terrestrial DON increase $NH_4^+$ concentration and nitrification increased $NO_2^-$ and $NO_3^-$ concentrations, while denitrification likely occurred on particle surface. Since nitrification exceeded denitrification, $NO_3^-$ addition was observed in the mixing zone. The riverine DIN discharge into the coastal ocean ranged from 77.2 t N day$^{-1}$ in the dry season and 101.5 t N day$^{-1}$ in the wet season. Due to the mild concentration of DIN in river water, Rajang-borne DIN likely adds positive effects in the coastal system, sustaining the primary productivity in the coastal zone. However, Rajang estuary is subject to intensive human development that frequently adds significant influences on DIN transport and transformation, which should receive further attention in the future study. In addition, the reaction trends and N solute distribution patterns obtained in the Rajang estuary could serve a reference for global N budget estimation.

**Author contribution**

JZ, MM, YW and SJ designed the study. JZ, EA, FJ and MM performed sample collection and in-situ measurement for the first cruise. SJ, KZ, AM, EA, FJ and MM performed sample collection and in-situ measurement for the second and the third cruise. SJ, JJ, GZ, YW, KZ, TR completed laboratory analyses. All co-authors equally participated in the interpretation and discussion of the results. SJ prepared the manuscript with suggestions from all co-authors.

**Competing interesting**

The authors declare that there is no conflict of interesting.

**Acknowledgements**

The present research was kindly supported by the Newton-Ungku Omar Fund (NE/P020283/1), China Postdoctoral Science Foundation (2018M630416), MOHE FRGS 15 Grant (FRGS/1/2015/WAB08/SWIN/02/1), Overseas Expertise Introduction Project for Discipline Innovation (B08022), SKLEC Open Research Fund (SKLEC-KF201610) and Scientific Research Foundation of SKLEC (2017RCDW04). The authors would like to thank the Sarawak Forestry Department and Sarawak Biodiversity Centre for permission to conduct collaborative research in Sarawak waters under permit numbers NPW.907.4.4(Jld.14)-161, Park Permit No WL83/2017, and SBC-RA-0097-MM. Lukas Chin and the "SeaWonder" crew are acknowledged for their support during the cruises. Technical support by Dr. Patrick Martin and Dr. Gonzalo Carrasco at Nanyang Technological University during the cruises and Ms. Lijun Qu, Ms. Wanwan Cao, Ms. Xiaohui Zhang, Dr. Xunchi Zhu at East China Normal University in the laboratory analyses is gratefully acknowledged. We also appreciate the great assistance from Prof. Zhiming Yu at the Institute of Oceanology, Chinese Academy of Sciences for the stable isotope analyses.



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





**Figure 1: Maps of the sampling area: (A) shows the location of the Rajang lower stream in the Borneo, which is invoked from Staub et al. (2000); (B) highlights the sampling sites in the September 2017 cruise; (C) outlines the distribution of peatland in the Rajang lower steam and coastal region (dark green) and deforestation documented in 2010 (Sarawak Geoportal: www.bmfmaps.ch). The cyan line indicates the Rajang River. The red dots represent Sarikei and Sibu. Both are important cities in the Rajang watershed.**

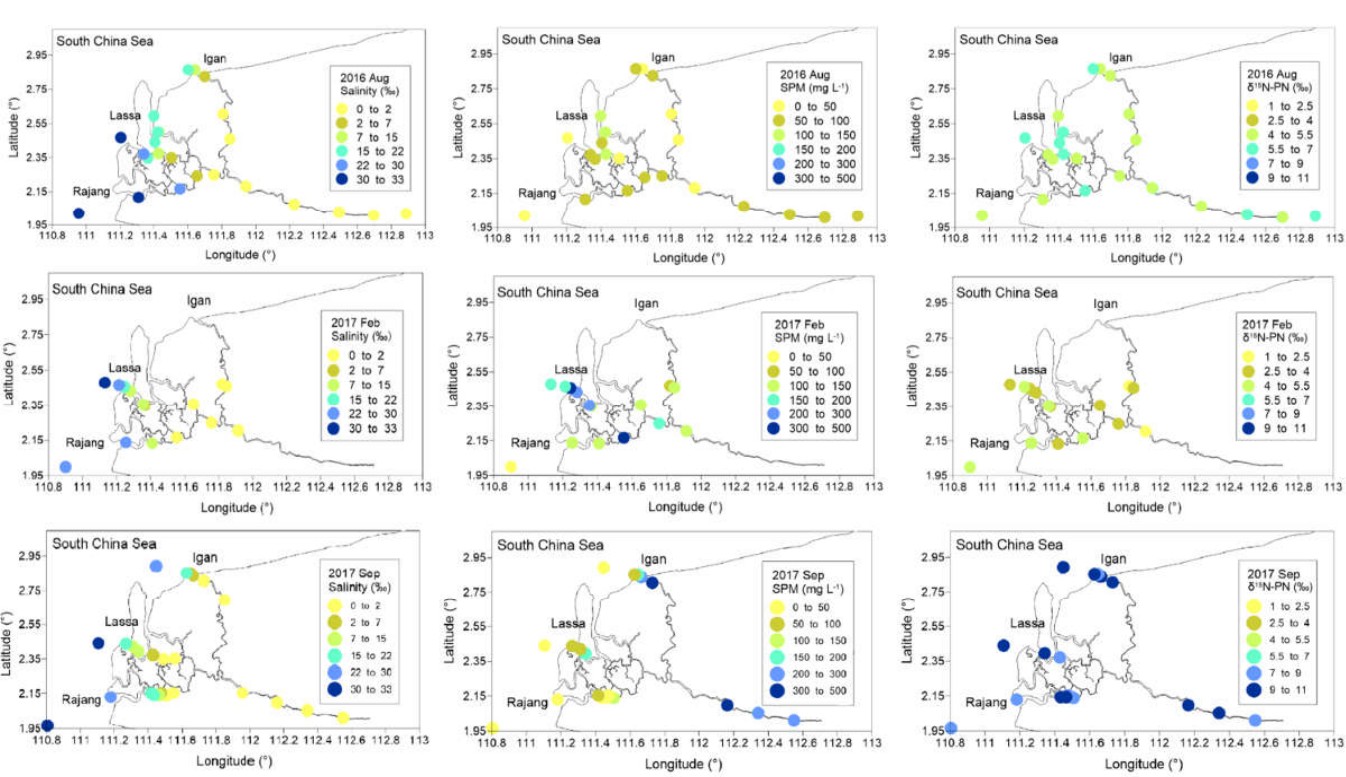

**Figure 2: Distribution of salinity, suspended particle matter (SPM) and isotope fraction δ15N-PN in the Rajang Estuary and adjacent coastal ocean.**



**Figure 3: Concentrations of NH₄⁺ and NO₃⁻ in the Rajang Estuary and adjacent coastal ocean; Distribution of δ¹⁵N-NO₃⁻ and δ¹⁸O-NO₃⁻ in the Rajang Estuary and adjacent coastal ocean.**



**Figure 4: Distribution of NH$_4^+$ and NO$_3^-$ concentrations, as well as δ$^{15}$N-NO$_3^-$, along the salinity gradient in the Rajang Estuary Concentrations of NH$_4^+$ and NO$_3^-$ in the Rajang Estuary and adjacent coastal ocean.**



**Figure 5: Distribution of NO$_3^-$ concentration offset ([ΔNO$_3^-$] in the figure) in three Rajang tributaries (Igan, Lassa and Rajang) along the salinity gradient from the three cruises; Distribution of δ$^{15}$N-NO$_3^-$ offset (Δδ$^{15}$N-NO$_3^-$ in the figure) in three Rajang tributaries along the salinity gradient in 2017; Comparison between δ$^{15}$N/$^{18}$O-NO$_3^-$ offset and NO$_3^-$ concentration offset for the cruises in 2017; Correlation between δ$^{15}$N-NO$_3^-$ and $^{18}$O-NO$_3^-$ offset, the slope of the correlation curve between δ$^{15}$N-NO$_3^-$ and δ$^{18}$O-NO$_3^-$ was 0.99 (R$^2$: 0.64).**





**Figure 6:  Concentration of DON, NO$_3$⁻, NH$_4$⁺ and NO$_2$⁻, as well as δ$^{15}$N-NO$_3$⁻ and δ$^{18}$O-NO$_3$⁻ in the mixing experiment (mean ± standard deviation).**





**Figure 7: (A)** Mean concentration of different N species in the fresh Rajang water (salinity: 0, sampling sites ranged from upper stream to Sibu city); **(B)** Comparison between $\delta^{15}N$-$NO_3^-$ and $\delta^{18}O$-$NO_3^-$ in the fresh Rajang water. The figure also highlights the range of both isotope fractions in rainwater, fertilizer, terrestrial organic matter and sewage (Li et al. 2010); **(C)** Comparison of DIN yield among different tropical rivers on the global scale; **(D)** Annual precipitation curve derived from (Müller et al. 2016) in Sarawak and precipitation in the sampling months during 2016 and 2017, measured at Sibu station. The figure also shows the Ocean Niño Index in 2016 and 2017. Clearly, in the beginning of 2016, an El Niño event occurred because the index was much higher than the threshold (red dash line: 0.5). Afterwards, continuous La Niña events were observed (lower than the threshold, highlighted by the blue dash line); **(E)** and **(F)** are comparisons among offsets (Δ in the figure) for $NO_3^-$, $NH_4^+$ and DON concentrations in the Lassa branch. The arrows outlined in the figure highlights the possible reaction pathways according to the variations in the offsets.





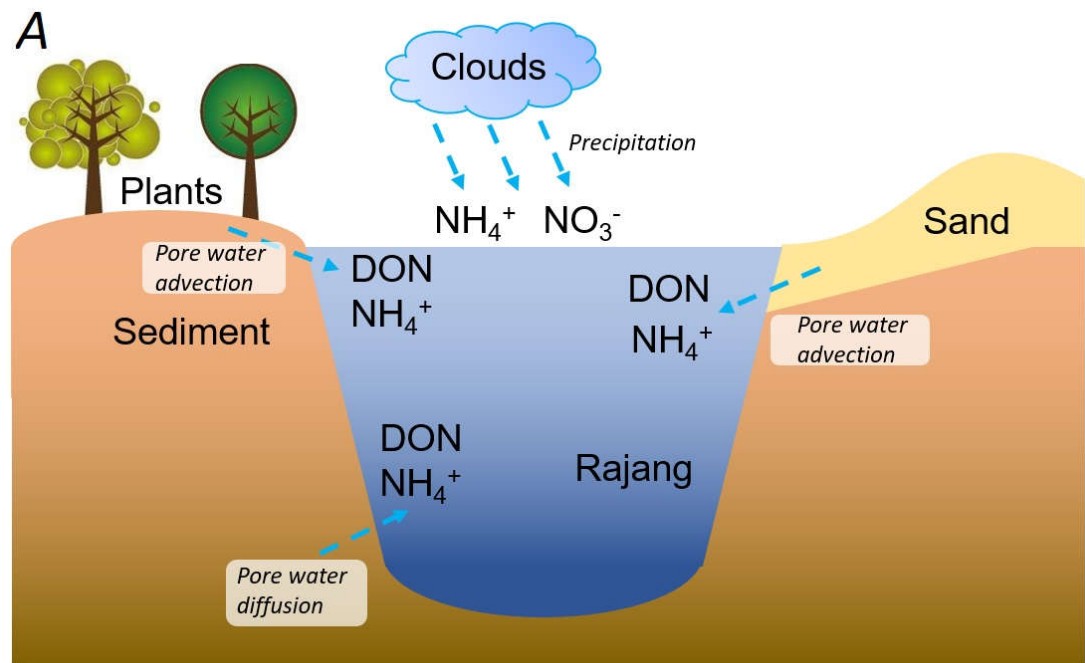

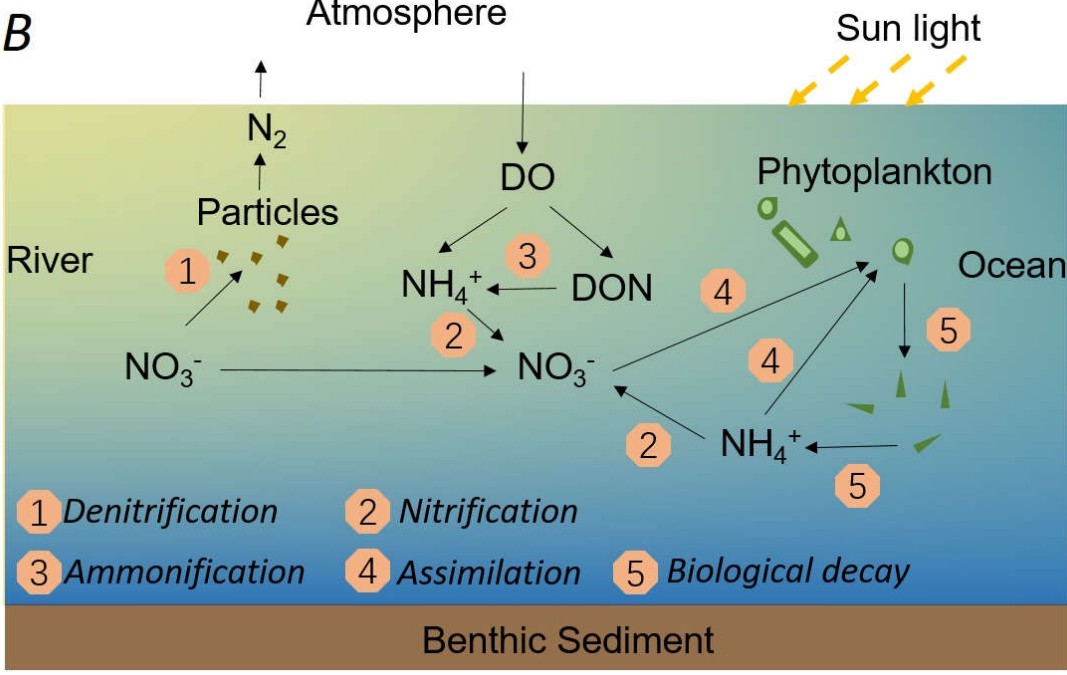

**Figure 8: Sketch of the N input pathways in the Rajang estuary (A) and N reactions during the mixing between river water and coastal ocean water obtained in the current research (B).**



**Table 1: Chemical properties in the rainwater and pore water from mangrove swamp, sandy beach and peatland.**

| Sample | Salinity (‰) | $NH_4^+$ (µM) | $NO_2^-$ (µM) | $NO_3^-$ (µM) | $\delta^{15}N\text{-}NO_3^-$ (‰) | $\delta^{18}O\text{-}NO_3^-$ (‰) | DON (µM) |
|---|---|---|---|---|---|---|---|
| Rainwater | 0 | 18.9 | 0.05 | 16.4 | 5.2 | 55.3 | 22.3 |
| Mangrove | 21.5 | 33.4 | 0.36 | 1.0 | 3.7 | 2.6 | 46.2 |
| Sandy beach | 17.5 | 22.8 | 0.05 | 0.3 | --- | --- | 34.5 |
| Peatland | 1.0 | 121 | 0.63 | 1.2 | 3.8 | 2.1 | 89.2 |

**Table 2: A global view on the $NH_4^+$ and $NO_3^-$ concentration in the fresh river water and the magnitude of riverine DIN flux. # Dry season record  * Monsoon Season record**

| Site | $NH_4^+$ (µM) | $NO_{2+3}^-$ (µM) | DIN flux (t N d⁻¹) | Reference |
|---|---|---|---|---|
| Rajang River (Malaysia)[#] | 5.5 | 5.4 | 77.2 | This study |
| Mekong River (Vietnam) | 11 | 15 | 789 | Lida et al., 2007 |
| | | | | Liljeström et al., 2012 |
| Danshui River (Taiwan, China) | 137 | 57.1 | 49.1 | Kuo et al., 2017 |
| Pearl River (China) | 207 | 211 | 468 | Cai et al., 2015 |
| Amazon River (Brazil) | 0.2 | 11.8 | 2322 | Santos et al., 2008 |
| Mississippi River (U.S.A) | 0.9 | 113 | 3720 | Battaglin et al., 2010 |
| Meghna River (Bangladesh)[*] | 0.1 | 15.8 | 85.0 | Uddin et al., 2014 |
| Pangani River (Tanzania) | 4.5 | 42.8 | 0.87 | Selemani et al., 2018 |
| Tana River (Kenya) | 0.62 | 9.6 | 1.57 | Bouillon et al., 2009 |
| Wanquan River (China) | 21 | 157 | 10.2 | Li et al., 2013 |