# Peer review of "Dissolved inorganic nitrogen in a tropical estuary at Malaysia: transport and transformation"

_Biogeosciences, 2019_

## Referee Comment (RC1) · Anonymous Referee #1 · 24 Mar 2019

Currently, the research documents for nitrogen circling in tropical estuaries are relatively limited, especially in South-eastern Asia. In the manuscript, the authors established a solid database, including dissolved nitrogen concentrations, particle nitrogen concentrations, and stable isotope fractions for nitrate and particle nitrogen, from three surveys in a tropical river, i.e. Rajang, in Sarawak state, Borneo. The authors described the source of dissolved inorganic nitrogen in the river water and likely reactions in the estuary. The seasonal variations and linkage between climate events and reactions were also included in the manuscript. This work adds valuable knowledge for us to understand the nitrogen turnover in the tropical zone. In addition, this manuscript provides useful information for nitrogen budget estimation on a global scale. The suggestions for the improvement of this manuscript are as follows. It would be glad to see

the revised version based on my feedback. Major comments: 1. In the introduction part, the authors introduced well about the other nitrogen research and the purpose in this paper. However, the meaning of this study was not quite clear. Maybe, for instance, 'the rapid industrialization of tropical countries had increased the N input from river which potentially increased the marine N load. This budget should be reevaluated precisely.' Please state the importance of this research and perhaps use some numbers to make the significance more attractive. 2. In the discussion part 2, authors indicated 'The overlying water seeped into the sediment along the conduits created by crabs and worms'. This process would probably modify the isotopic compositions of nitrate according to some research, which might affect the isotope analysis of nitrogen transformation. Is this phenomenon can be discovered in this study?

Minor comments: 1. Page 1, line 29, change 'Gruber and Galloway 2008' to 'Gruber and Galloway, 2008) 2. Page 2, line 12, change 'isotope ratios' into 'isotope fractions' because $\delta$ indicates the fraction. 3. Page 2, line 15, 'Yan et al. 2007' should be '2017' 4. Page 2, line 26, it is better to be 'response . . . to. . .', not 'response. . . on. . .' 5. Page 2, line 32, it should be 'typically tropical' 6. In the MM section, authors described the figure as 'Fig. 1 B', while it was changed to 'Fig. 7A' in the Discussion, please keep one style. 7. Page 3, line 7, add space in 'Fig.1' 8. Page 3, line 15, please change to 'Rajang is the largest river' 9. Page 3, line 20, change 'accumulation for 6000 years' to '6000-year accumulation' 10. Page 3, line 25, change 'tributaries' to 'tributary' 11. Page 3, line 26, change 'meter' to 'meters' or 'm' 12. Page 4, line 4, authors used U.S. here and used 'USA' in the following, please change to 'USA' 13. Page 4, line 7, I believe authors did not collect pore water before Sep 2017, so remove 'Additional' in this sentence 14. Page 4, line 10, change 'HDPE' to 'High Density Polyethylene (HDPE)' 15. Page 4, line 12, please define 'DO' here 16. Page 4, line 13, perhaps it is better with 'at low tide' 17. Page 6, line 8, it is better to offer the number of figure in SM. You can add '(Fig. S1 to Fig. S10) at the end of the sentence. 18. Page 5, line 10, the definition of DON has been mentioned on page 2, you can use DON directly 19. Page 8, line 4, maybe it is better to use 'Rajang river water' compared to 'fresh river water'

20. Page 8, line 7, authors used U.S.A. here, please be unified. In addition, add 'the' in front of 'Pearl River', the same for the remaining rivers in the sentence. 21. Page 9, line 7, change to 'This concentration increase generally results from...' 22. Page 9, line 8, add 'the' before 'Rajang Delta' 23. Page 9, line 9, the streams also can be found from Fig. 1. Maybe it is better to add Fig. 1 as a reference as well. 24. Page 9, line 13, it is better to be 'large contact areas' 25. Page 9, line 14, maybe wave actions and density difference also enhance the pore water exchange. Authors can add these effects or add 'mainly' in the sentence. 26. Page 9, line 30, it is better to be '15N-NO3-enriched water, i.e..., into Rajang.' 27. Page 9, line 31, 'reduction in DON', did you mean 'compared to the conservative mixing'? Please clarify. 28. Page 10, line 27, it is better to be 'significant level' than 'identifiable level' 29. Page 10, line 28, passive voice is necessary. 'The influence... may be enhanced.' 30. Page 11, line 31, it is better to be 'along several streams' 31. Page 12, line 6, authors used italic f here, please change the characters in other sections at Discussion 32. Page 12, line 27, authors used 'receive more attention' several times, maybe change to 'be noticed' 33. Page 13, line 1, I think it not proper to say the linkage between climate and N circling, please use climate events instead 34. Page 13, line 3, please change to 'on particle surfaces' 35. Page 13, line 9, I believe it should be passive voice, please change to 'can be treated as a...' 36. Page 13, line 13, it should be 'for the second and third cruise' 37. Page 13, line 13 and 14, please set 'in-situ' in italics. 38. Page 15, line 17, pleased remove 'Suimon Mizu Shigen Gakkaishi' from the reference, I think it is an error from authors' software 39. Page 17, line 7, it looks like the authors missed an author (B. D. Eyre) for the reference 'The driving forces of porewater and groundwater flow in permeable coastal sediments: A review' 40. Figure 1, the word in Fig. 1C is a bit blur. In addition, the authors did not clarify the meaning for the yellow color in Fig. 1B (I guess it is deforestation) in the legend. Please add this information. 41. Figure 7, the reference style should be changed in the legend 42. Figure 8, please separate A and B a bit, it is easy to misunderstand 'atmosphere'

---

## Referee Comment (RC2) · Anonymous Referee #2 · 1 Apr 2019

Dissolved inorganic nitrogen in a tropical estuary at Malaysia: transport and transformation. Authors: Shan Jiang et al.

Increasing nitrogen enrichment is one of the main pressures compromising the integrity of coastal ecosystems. Given the rapidly changing tropic coastal areas, the present paper is timely and an important contribution towards a better understanding of a rapidly changing global nitrogen cycle. The authors describe a series of cruises in the Rajang estuary where they investigated the distribution of nitrogen species including 15N/18O Isotopes. By combining observed distribution patterns with a dedicated series of incubations, the authors derive estimates of N transformation, and based on this, an improved estimate of riverine nitrogen loads by the Rajang river to the coastal ocean.

In general, the paper is well written and most results are clearly presented. However,

the discussion is quite long with several unclear sentences: Itneeds to be more focused and needs attention in terms of clarity. On many instances, claims are made that are not backed up by literature or data. I suggest to focus on a few points for which clear cases are presented.

The title focusses on inorganic nitrogen but dissolved organic nitrogen is playing a crucial role. The authors may consider to leave "inorganic" from the title.

The language was mostly clear, but it is important that the text is corrected by a native speaker. E.g. often articles are missing.

Suggestions and questions Abstract, Line 17 – 19. Split sentence into two: La Niña induced high precipitation and discharge rates, decreased reaction intensities of ammonification and nitrification. Hence similar distribution patterns of DIN species in the estuary were found during both seasons.

Page 5, Line 7: Precision instead of pression

Page 6, line 15. I do not agree that the concept of Apparent Oxygen Utilization can be applied as the river is an open system. Hence, an unknown amount of O2 is exchanged with the atmosphere. I strongly suggest to use undersaturation instead.

Page 6, line 20/21: In most estuarine literature, this phenomenon is referred to as an estuarine turbidity maximum. I suggest to use that term. What was the SPM concentration in the sea?

Page 6, line 24: the correlation with salinity is not evident from S4. If that correlation is not important, I suggest to delete/reword the sentence of alternatively show whether the correlation is significant.

Page 7, line 1. PN values are given in mg/l, dissolved fractions in mol/l. I suggest to convert the PN also in mol to simplify a comparison.

Page 7, line 4. Leaving out which data? And why? Are they shown somewhere? What

is the effect on the conclusions?

Page 7, line 13. What was meant: a deviation from linear mixing? Please be more precise (You explained the principle in material and methods). E.g. in addition to offset name it "deviation from conservative mixing" the first time you use the concept. And mention that NO3 is released.

Page 7, line 19. Conservative mixing instead of …. distribution?

Page 8, line 1. Porewater samples was limited? Maybe you mean low? How low?

Page 8, line 5 I am not familiar with using "aorta" to refer to river characteristics. Please clarify/use other terms. Page 8, line 30: ….was comparably "wet" than… : unclear sentence: wetter than ?

Page 8. Section 4.1: Do the stable isotopes support that higher N release in Rajang watershed is from fertilizers? This is not apparent from Fig 7. Alternatively, it is related to degradation of peatlands. I suggest the authors to improve their case(s) in Section 4.1. Specifically, it would help if the authors are able to discern between two important sources fertilizer/human sources and N from the oxidation of peat. Can the stable isotopes help? Also, I suggest to discern more clearly between increased loads due to increased runoff (la nina) and due to increased concentrations.

Page 9. Section 4.2: First paragraph How much pore water exchange is necessary to explain the observed increase? Is this realistic or are other processes be involved?

Page 9. Section 4.2: second paragraph You claim that PN is not involved in the transformation processes, but given the high PN concentrations and low DIN concentrations, small changes in PN may have a large impact on DIN. I suggest to do some simple calculations, how much PN has to be reduced to explain the observed DIN changes. See also comment to page 10, line 33ff

Page 9, line 32/33. This sentence reads as if DON in the mixing zone is lower than in the coastal ocean. Please rephrase.

Page 10. Line 10:... their input can be identified. This is just a claim. Please substantiate.

Page 10, line 20 – 28. This part of the discussion could fit better in Part 4.1

Page 10, line 33 ff. Here you claim that PN can play a role, but in 4.1 you claim that PN does not play a role. Please clarify this.

Page 11. Line 6 ff. No supporting parameters like chlorophyll are presented that may clarify changes in PN quality. In this respect, I wonder about whether phytoplankton blooms occur? After all, fresh readily degradable organic matter is needed to create the anoxic microniches needed for denitrification. Please clarify this. Also, can you discern between sediment denitrification and water column denitrification?

Page 25, line 25. This observation reinforced.........: please add a citation to back this statement.

Section 4.3 Formula (4) should be transferred to the Material and methods section.

Of course, the total loads are strongly dependent on discharge. For that reason I suggest no to focus on loads but on the concentrations: What are the factors responsible for the observed rather low DIN loads???

Discussion: General Comment In general, the discussion is too long. The points addressed in the discussion are important. But the paper would gain, if the discussion is more focused than at present.

---

## Author Comment (AC1) · 27 Apr 2019

Responses to Referees

1st Referee

Major comments:
1. In the introduction part, the authors introduced well about the other nitrogen research and the purpose in this paper. However, the meaning of this study was not quite clear. Maybe, for instance, 'the rapid industrialization of tropical countries had increased the N input from river which potentially increased the marine N load. This budget should be reevaluated precisely.' Please state the importance of this research and perhaps use some numbers to make the significance more attractive.
Reply: Thank you. Indeed, the importance of the current research should be better highlighted. In the revised manuscript, we modified some sentences in the introduction by adding numbers and descriptions.
Revised manuscript: Page 2: The ecological and environmental response in tropical estuaries to the DIN related anthropogenic pressure is less documented and the magnitude of river borne DIN fluxes to the coastal line in tropical zones is less evaluated.
Page 2: The tropical zone, characterized by the high annual temperature and intensive precipitations, hosts substantial rivers and streams; while the studies on DIN transport are scarce, especially in Southeast Asia.
Page 2: In addition, since the Second World War, a rapid development in tropical countries has been witnessed. For instance, from 1960 to 2008, the gross national income (GNI) per capita in Malaysia (3.3-5.3) is much higher than the global average level (0.7-1.8; Tran, 2013). Coupled with urbanization, land use change and population increasing, the tropical zone is becoming a hotspot for DIN production, utilization and transport (Seitzinger et al., 2002).

2. In the discussion part 2, authors indicated 'The overlying water seeped into the sediment along the conduits created by crabs and worms'. This process would probably modify the isotopic compositions of nitrate according to some research, which might affect the isotope analysis of nitrogen transformation. Is this phenomenon can be discovered in this study?
Reply: Thank you. Indeed, the pore water borne nitrate could directly change the isotope fractions in the receiving water. In the current research, the nitrate concentration was low in all the pore water samples, as outlined in Table 1. Consequently, the direct influence on the $\delta^{15}N$-$NO_3^-$ and $\delta^{18}O$-$NO_3^-$ is limited. The indirect influence caused by $NH_4^+$ injection might be significant because of high concentrations in the pore water. The pore water derived $NH_4^+$ and mineralized $NH_4^+$ were mixed in the river water. We observed the decline in isotope fractions, likely due to nitrification. However, the sole effect from pore water was difficult to be identified in the present study. In the revised manuscript, we modified our expressions, highlighting the direct influence was minor and the nitrification in the river partly relies on pore water derived $NH_4^+$.
Revised manuscript: Page 10: Notably, low $NO_3^-$ levels were found in pore water samples, suggesting that the direct effect on the river water NO3- concentration and its isotope fractions was limited.
Page 10: The elevation in the concentrations of $NO_2^-$ and $NO_3^-$ was attributed to nitrification,

relying on the mineralized $NH_4^+$ and pore water derived $NH_4^+$ (Fig. 8B)

Minor comments:
1. Page 1, line 29, change 'Gruber and Galloway 2008' to 'Gruber and Galloway, 2008)
Reply: Thank you. We have added a comma.
Revised manuscript: Consequently, DIN is assumed to be the most active component in the N cycle and the transport of DIN among ecosystems is a hotspot in biogeochemical research on a global scale (Gruber and Galloway, 2008).

2. Page 2, line 12, change 'isotope ratios' into 'isotope fractions' because δ indicates the fraction.
Reply: Thank you. We have changed it.
Revised manuscript: More importantly, stable isotope fractions, e.g. $\delta^{15}N\text{-}NO_3^-$, $\delta^{18}O\text{-}NO_3^-$, $^{15}N\text{-}$PN (particle nitrogen), have been introduced to trace DIN transport and transformation processes (Middelburg and Nieuwenhuize, 2001).

3. Page 2, line 15, 'Yan et al. 2007' should be '2017'
Reply: Thank you. We have changed it.
Revised manuscript: The stable isotope technique has been applied in a number of estuaries located in temperate and sub-tropical zones, such as the Changjiang estuary (China; Yu et al., 2015; Yan et al., 2017)…

4. Page 2, line 26, it is better to be 'response : : : to: : :', not 'response: : : on: : :'
Reply: Thank you. We have changed it.
Revised manuscript: The ecological and environmental response in tropical estuaries to the DIN related anthropogenic pressure is less documented.

5. Page 2, line 32, it should be 'typically tropical'
Reply: Thank you. We have changed it.
Revised manuscript: In the present study, three cruises in a typically tropical estuary.

6. In the MM section, authors described the figure as 'Fig. 1 B', while it was changed to 'Fig. 7A' in the Discussion, please keep one style.
Reply: Thank you. We have unified all the formats in the revised manuscript.
Revised manuscript: The Rajang river (hereafter Rajang) is located in Sarawak state, Borneo (Fig. 1A and B).

7. Page 3, line 7, add space in 'Fig.1'
Reply: Thank you. We have added a space.
Revised manuscript: The Rajang river (hereafter Rajang) is located in Sarawak state, Borneo (Fig. 1A and B).

8. Page 3, line 15, please change to 'Rajang is the largest river'
Reply: Thank you. We have changed it.
Revised manuscript: The Rajang is the largest river in Malaysia with a length of ca. 530 km.

9. Page 3, line 20, change 'accumulation for 6000 years' to '6000-year accumulation'

Reply: Thank you. We have changed it.

Revised manuscript: After 6000-year accumulation, sediment particles from the upper stream developed a large area of alluvial delta (from Sibu to estuary mouths).

10. Page 3, line 25, change 'tributaries' to 'tributary'

Reply: Thank you. We have changed it.

Revised manuscript: Apart from the Rajang tributary, the peat deposits were frequently observed in the remaining three tributaries (Fig. 1C).

11. Page 3, line 26, change 'meter' to 'meters' or 'm'

Reply: Thank you. We have changed to 'm'.

Revised manuscript: The tidal range in the Rajang estuary is from 2.8 to 5.6 m and decreases from the Rajang (macro tidal range) to the Igan (meso tidal range).

12. Page 6, line 4, authors used U.S. here and used 'USA' in the following, please change to 'USA'

Reply: Thank you. We have unified all expressions to 'USA'

Revised manuscript: The spatial distribution of each parameter, e.g. salinity, temperature and solute concentration, was plotted in Surfer 14.0 (Golden Software Inc., USA) and the dot plots were done in Sigmaplot 12.5 (Systat Software Inc., USA).

13. Page 4, line 7, I believe authors did not collect pore water before Sep 2017, so remove 'Additional' in this sentence

Reply: Thank you. We have removed it.

Revised manuscript: In September 2017, pore water samples from the edge of mangrove forest, peatland and coastal sandy beach were collected. Rainwater was gathered once during the September cruise.

14. Page 4, line 10, change 'HDPE' to 'High Density Polyethylene (HDPE)'

Reply: Thank you. We have changed it.

Revised manuscript: The river water and coastal seawater were collected into 1 L acid-prewashed high density polyethylene (HDPE) sampling bottles via a pole-sampler that decreases the contamination from boat surface and engine cooling water (Zhang et al., 2015).

15. Page 4, line 12, please define 'DO' here

Reply: Thank you. In the revised manuscript, the definition of DO was provide at the beginning of the Introduction section (Second paragraph). Therefore, we did not add explanation here.

16. Page 4, line 13, perhaps it is better with 'at low tide'

Reply: Thank you. We have changed it.

Revised manuscript: For pore water samples, a sampling hole with a depth of approximately 20 cm was dug at low tide.

17. Page 6, line 8, it is better to offer the number of figure in SM. You can add '(Fig. S1 to Fig. S10) at the end of the sentence.
Reply: Thank you. We have added this information.
Revised manuscript: The spatial distribution of each parameter, e.g. salinity, temperature and solute concentration, was plotted in Surfer 14.0 (Golden Software Inc., USA) and the dot plots were done in Sigmaplot 12.5 (Systat Software Inc., USA). Given the limited space, a portion of plots was displayed in supplementary materials (Fig. S1 to Fig. S10).

18. Page 5, line 10, the definition of DON has been mentioned on page 2, you can use DON directly
Reply: Thank you. We have changed it.
Revised manuscript: The difference in the concentration between TDN and DIN was assumed to be the level of DON.

19. Page 8, line 4, maybe it is better to use 'Rajang river water' compared to 'fresh river water'
Reply: Thank you. We have changed it.
Revised manuscript: DIN sources in the Rajang river water.

20. Page 8, line 7, authors used U.S.A. here, please be unified. In addition, add 'the' in front of 'Pearl River', the same for the remaining rivers in the sentence.
Reply: Thank you. We have changed it and added 'the' in the sentence.
Revised manuscript: In comparison to rivers located in dense population regions, such as the Pearl River in China, the Mississippi River in the USA, the Danshui River in Taiwan, China, and the Mekong River in Vietnam, concentrations of $NO_3^-$ and $NH_4^+$ in the Rajang water were low (Table 2).

21. Page 9, line 7, change to 'This concentration increase generally results from: : :'
Reply: Thank you. We have changed it.
Revised manuscript: This concentration increase generally results from (1) a direct input from tributary streams or pore water exchange and (2) N transformations.

22. Page 9, line 8, add 'the' before 'Rajang Delta'
Reply: Thank you. We have added 'the' in the manuscript.
Revised manuscript: In the Rajang Delta, there are several small streams (Fig. 1), continuously adding solutes into the estuary water (Staub et al., 2000; Gaslto, 2010).

23. Page 9, line 9, the streams also can be found from Fig. 1. Maybe it is better to add Fig. 1 as a reference as well.
Reply: Thank you. We have added 'Fig. 1' in the sentence to support the presence of streams in the Rajang Delta.
Revised manuscript: In the Rajang Delta, there are several small streams (Fig. 1), continuously adding solutes into the estuary water (Staub et al., 2000; Gaslto, 2010).

24. Page 9, line 13, it is better to be 'large contact areas'

Reply: Thank you. We have changed it.

Revised manuscript: Alternatively, the exchange between pore water in both cohesive and sandy sediments and surface water can add DIN in river water on a regional scale due to large contact areas (Fig. 8A).

25. Page 9, line 14, maybe wave actions and density difference also enhance the pore water exchange. Authors can add these effects or add 'mainly' in the sentence.

Reply: Thank you. We have added 'wave actions' and 'density difference' in the sentence.

Revised manuscript: In coastal areas, this exchange can be driven by tidal pumping, wave actions and density difference (Santos et al., 2012a)

26. Page 9, line 30, it is better to be '15N-NO3-enriched water, i.e: : :, into Rajang.'

Reply: Thank you. We believe this comment belongs to Page 11, line 30. In the revised manuscript, we have changed the sentence.

Revised manuscript: The anthropogenic activity likely introduced $^{15}$N- $NO_3^-$ enriched water, i.e. wastewater or sewage, into the Rajang (Fig. 7B), along several streams.

27. Page 9, line 31, 'reduction in DON', did you mean 'compared to the conservative mixing'? Please clarify.

Reply: Indeed, compared with the conservative distribution, a reduction in DON content was observed. In the revised manuscript, we have added the information 'compared to the conservative mixing'.

Revised manuscript: In the mixing experiment, a reduction in DON concentration was observed, compared to the conservative mixing, which confirms the biogeochemical activity.

28. Page 10, line 27, it is better to be 'significant level' than 'identifiable level'

Reply: Thank you. We have changed 'identifiable level' to 'significant level'.

Revised manuscript: The influence may not reach a significant level.

29. Page 10, line 28, passive voice is necessary. 'The influence: : : may be enhanced.'

Reply: Thank you. We have changed the sentence into passive voice.

Revised manuscript: However, disturbances from the shrinkage of peatland may be enhanced in the future.

30. Page 11, line 31, it is better to be 'along several streams'

Reply: Thank you. We have removed 'along' in the sentence.

Revised manuscript: Anthropogenic activities likely introduced $^{15}$N- $NO_3^-$ enriched water, i.e. wastewater or sewage, into the Rajang (Fig. 7B), along several streams.

31. Page 12, line 6, authors used italic f here, please change the characters in other sections at Discussion

Reply: Thank you. We have changed all 'f' in the sentence, using italics.

32. Page 12, line 27, authors used 'receive more attention' several times, maybe change to 'be noticed'

Reply: Thank you. We have changed it.

Revised manuscript: It has increased suspended particle concentration and accelerates erosion of terrestrial organic matter (Ling et al., 2017), indicating that a long-term influence derived from the human transformations should be noticed in future studies.

33. Page 13, line 1, I think it not proper to say the linkage between climate and N circling, please use climate events instead

Reply: Thank you. We have changed it.

Revised manuscript: This indicates a causal chain between climate events and N cycling in tropical soils and rivers.

34. Page 13, line 3, please change to 'on particle surfaces'

Reply: Thank you. We have changed it.

Revised manuscript: In the estuary mixing zone, pore water exchange and decomposition of terrestrial DON increase $NH_4^+$ concentration and nitrification increased $NO_2^-$ and $NO_3^-$ concentrations, while denitrification likely occurred on particle surfaces.

35. Page 13, line 9, I believe it should be passive voice, please change to 'can be treated as a: : :'

Reply: Thank you. We have changed the sentence in the revised manuscript.

Revised manuscript: In addition, the reaction trends and N solute distributions obtained from the Rajang estuary may benefit global N budget estimation.

36. Page 13, line 13, it should be 'for the second and third cruise'

Reply: Thank you. We have changed it.

Revised manuscript: SJ, KZ, AM, EA, FJ and MM performed sample collection and *in-situ* measurement for the second and third cruise.

37. Page 13, line 13 and 14, please set 'in-situ' in italics.

Reply: Thank you. We have modified it.

Revised manuscript: JZ, EA, FJ and MM performed the sample collection and *in-situ* measurements for the first cruise. SJ, KZ, AM, EA, FJ and MM performed samplings and *in-situ* measurements for the second and third cruise.

38. Page 15, line 17, pleased remove 'Suimon Mizu Shigen Gakkaishi' from the reference, I think it is an error from authors' software

Reply: Thank you. We have removed it.

Revised manuscript: Iida, T., Inkhamseng, S., Yoshida, K. and Ito, S.: Seasonal Variation in Nitrogen and Phosphorus Concentrations in the Mekong River at Vientiane. Journal of Japan Society of Hydrology & Water Resources, 20, 226-234, 2007.

39. Page 17, line 7, it looks like the authors missed an author (B. D. Eyre) for the reference 'The driving forces of porewater and groundwater flow in permeable coastal sediments: A

review'

Reply: Thank you. We have added it.

Revised manuscript: Santos, I. R., Eyre, B. D. and Huettel, M.: The driving forces of porewater and groundwater flow in permeable coastal sediments: A review, Estuar. Coast Shelf S., 98, 1-15, 2012.

40. Figure 1, the word in Fig. 1C is a bit blur. In addition, the authors did not clarify the meaning for the yellow color in Fig. 1B (I guess it is deforestation) in the legend. Please add this information.

Reply: Thank you. We have highlighted that yellow color represents deforestation in the manuscript. We also retyped the names of two cities, Sibu and Sarikei.

41. Figure 7, the reference style should be changed in the legend

Reply: Thank you, we have changed 'et al.' to 'et al.,' in the legend.

42. Figure 8, please separate A and B a bit, it is easy to misunderstand 'atmosphere'

Reply: Thank you, we have moved B and separated A and B.

2nd Referee

In general, the paper is well written and most results are clearly presented. However the discussion is quite long with several unclear sentences: It needs to be more focused and needs attention in terms of clarity. On many instances, claims are made that are not backed up by literature or data. I suggest to focus on a few points for which clear cases are presented.

Reply: Thank you for reviewing the manuscript. In the revised draft, we made several changes in the discussion. We added more information in the N circling in the estuary. For the flux section, we reduced the passage by moving the calculation to MM section and deleting some descriptions in order to highlight the key points in the manuscript.

The title focusses on inorganic nitrogen but dissolved organic nitrogen is playing a crucial role. The authors may consider to leave "inorganic" from the title.

Reply: Thank you. Indeed, this draft invokes organic N and particle N. We discussed whether we should use 'nitrogen' or 'dissolved nitrogen' in the title. Eventually, we chose 'dissolved inorganic nitrogen' to highlight the key of this manuscript. Dissolved organic nitrogen and particle nitrogen are the reference to support the transformations of inorganic fraction in the estuary.

The language was mostly clear, but it is important that the text is corrected by a native speaker. E.g. often articles are missing.

Reply: Thank you. The other reviewer also gives some suggestions on the article usage in the manuscript. In the revised draft, we tried to improve the grammar.

Abstract, Line 17 – 19. Split sentence into two: La Niña induced high precipitation and

discharge rates, decreased reaction intensities of ammonification and nitrification. Hence similar distribution patterns of DIN species in the estuary were found during both seasons.

Reply: Thank you. In the revised draft, we have modified the sentence according to this suggestion. Considering that the original sentence is the description of the trend in September 2017, we changed 'during both seasons' to 'in September 2017 (end of dry season)'.

Revised manuscript: La Niña induced high precipitations and discharge rates, decreased reaction intensities of ammonification and nitrification. Hence similar distribution patterns of DIN species in the estuary were found in September 2017 (end of dry season).

Page 5, Line 7: Precision instead of pression

Reply: Thank you. We have modified this typo.

Revised manuscript: The determination limit for these species was below 0.1 µM and analysis precision was 3.5%.

Page 6, line 15. I do not agree that the concept of Apparent Oxygen Utilization can be applied as the river is an open system. Hence, an unknown amount of O2 is exchanged with the atmosphere. I strongly suggest to use undersaturation instead.

Reply: Thank you. Indeed, the oxygen exchange between river water and atmosphere can be intensive. In the revised manuscript, we changed apparent oxygen utilization into oxygen undersaturation degree (OUD) through the manuscript. In addition, we modified the figures via changing AOU to OUD (Figure S4).

Revised manuscript: Accordingly, the oxygen undersaturation degree (OUD), i.e. the difference between DO saturation level and observed DO concentration, was calculated.

Page 6, line 20/21: In most estuarine literature, this phenomenon is referred to as an estuarine turbidity maximum. I suggest to use that term. What was the SPM concentration in the sea?

Reply: Thank you. We agree with you that it is usually referred as turbidity maximum. In the revised draft, we changed the expression. For SPM concentration in the sea, it was 24 mg L$^{-1}$ (the minimum in the range). In the revised draft, we highlighted this value.

Revised manuscript: Compared with the coastal ocean (24 mg L$^{-1}$), the SPM content in river water was markedly higher and presence of estuarine turbidity maximum was obtained in river branches.

Page 6, line 24: the correlation with salinity is not evident from S4. If that correlation is not important, I suggest to delete/reword the sentence of alternatively show whether the correlation is significant.

Reply: Thank you. There is no significant correlation between PN and salinity. In the revised manuscript, we removed this sentence.

Page 7, line 1. PN values are given in mg/l, dissolved fractions in mol/l. I suggest to convert the PN also in mol to simplify a comparison.

Reply: Thank you. In the revised manuscript and figures, we have changed the unit for PN into µmol L$^{-1}$ (µM).

Page 7, line 4. Leaving out which data? And why? Are they shown somewhere? What is the effect on the conclusions?

Reply: Thank you. Here, we did not leave any points for the discussion. We would like to express that the concentration of DON in majority of sites during and Feb 2017 and Sep 2017 survey was similar with the previous survey (Aug 2016). These data also showed in the figure (Fig. S7). All the conclusions and statements were made on the basis of the entire data. As you suggested, this sentence may lead to confusion to readers. We modified this sentence in the revised draft.

Revised manuscript: In the remaining two cruises, DON concentrations in many sites were comparable to the range obtained from the first cruise.

Page 7, line 13. What was meant: a deviation from linear mixing? Please be more precise (You explained the principle in material and methods). E.g. in addition to offset name it "deviation from conservative mixing" the first time you use the concept. And mention that NO3 is released.

Reply: Thank you. We tried several times to find 'a deviation from linear mixing'. However, it is not in Page 7, line 13, as the picture shown.

[Figure]

**3.3 Dissolved nitrogen and related isotope fractions**

Compared with PN concentration, the content of dissolved fractions was relatively minor. In August 2016, the DON concentration varied from 2.6 to 14.8 µM and high levels were found in estuary channels (Fig. S6). In the remaining two cruises, excluding several sites with high levels that were patchily distributed in tributaries in September 2017, a similar

5 concentration range was obtained. The $NH_4^+$ concentration was 0.39 to 17.3 µM (Fig. 3). Similar with DON, the seasonal variation in $NH_4^+$ concentration was limited. During the mixing, a slight increase in $NH_4^+$ concentration in August cruise was found (Fig. 4), especially in the Lassa tributary; while the remaining two cruises showed a limited variation during the river-ocean mixing. For $NO_3^-$ concentration in the river water, it varied between 1.6 and 14.8 µM in August 2016 (Fig. 3). The concentration in the remaining two cruises slightly decreased. The highest concentration was below the level of 10 µM. The

10 similarity derived from $NO_3^-$ concentration distribution in 2017 cruises led to an insignificant variation in the range for $\delta^{15}$N-$NO_3^-$ and $\delta^{18}$O-$NO_3^-$ between seasons (Fig. 3). In terms of $NO_2^-$, i.e. the minimum component in DIN inventory, the concentration varied between 0.09 and 3.3 µM in August 2016. Similar levels were found in the remaining samples.

Along the salinity gradient, positive offset for $NO_3^-$ concentration was frequently observed, especially in Lassa and Igan branches, which suggests net generations during the mixing (Fig. 5). Coupled with $NO_3^-$ generation, negative offsets in $\delta^{15}$N-

15 $NO_3^-$ and $\delta^{18}$O-$NO_3^-$ at most sites were observed (Fig. 5). The relationship between offsets of $\delta^{15}$N-$NO_3^-$ and $\delta^{18}$O-$NO_3^-$ were linearly correlated (Fig. 5). The slope was 0.99 and $R^2$ was 0.63 ($p<0.05$).

We also give a search on the word manuscript for several times. We cannot locate any place of 'a deviation from linear mixing'. After discussion, we believe you would like to mention the offset. In the revised manuscript, we added the explanation for the offset in the Results according to this suggestion.

Revised manuscript: Along the salinity gradient, positive offsets (positive deviation from conservative mixing) for $NO_3^-$ concentration were frequently observed, especially in Lassa and Igan branches, which suggests net generations ($NO_3^-$ release) during the mixing (Fig. 5).

Page 7, line 19. Conservative mixing instead of ∵∴. distribution?

Reply: Thank you. We have changed it.

Revised manuscript: In the experiment, DON concentration slightly departed from the conservative mixing.

Page 8, line 1. Porewater samples was limited? Maybe you mean low? How low?

Reply: Thank you. We wanted to express that nitrite and nitrate levels were low in pore water samples because the highest concentration was 1.2 μM. In the revised draft, we changed 'limited' to 'low' in the sentence and add the concentration.

Revised manuscript: In comparison, the level of $NO_3^-$ and $NO_2^-$ in pore water samples was low (<1.2 μM).

Page 8, line 5 I am not familiar with using "aorta" to refer to river characteristics. Please clarify/use other terms.

Reply: Thank you. We would like to express that the Rajang is the largest river in Sarawak. In the revised draft, we have changed this expression.

Revised manuscript: The Rajang is the largest river in Sarawak and receives substantial materials from its watershed.

Page 8, line 30: ∵∴.was comparably "wet" than∵∴ unclear sentence: wetter than ?

Reply: Thank you. Indeed, it should be wetter.

Revised manuscript: Consequently, the weather in September 2017 was comparably 'wetter' than the dry season in August 2016.

Page 8. Section 4.1: Do the stable isotopes support that higher N release in Rajang watershed is from fertilizers? This is not apparent from Fig 7. Alternatively, it is related to degradation of peatlands. I suggest the authors to improve their case(s) in Section 4.1. Specifically, it would help if the authors are able to discern between two important sources fertilizer/human sources and N from the oxidation of peat. Can the stable isotopes help? Also, I suggest to discern more clearly between increased loads due to increased runoff (la nina) and due to increased concentrations.

Reply: Thank you. In Fig. 7B, we used the plot of dual stable isotope fractions to identify the potential sources for nitrate in the river water (salinity: 0). It can be observed that points mainly fell in the range of fertilizer and soil organic N. This method is dependent on experiences (the range for $^{15}N$-$NO_3^-$ and $^{18}O$-$NO_3^-$). Consequently, it is difficult to pinpoint the exact source. In addition, the biogeochemical reactions may also alter the isotope fractions, leading to errors in the estimation. In the revised draft, we added this information to explain the potential disadvantage.

For the N from peat environment, the stable isotope could not effectively separate these processes in the current circumstance. In the revised manuscript, we highlighted the possibility for the presence of several sources.

For the effect of La Niña, the discharge rate increased while the $NH_4^+$ concentration decreased. The concentration of $NO_2^-$ and $NO_3^-$ was similar among cruises. In the revised manuscript, we considered the dilution effect for $NH_4^+$, and offered more references to explain the linkage

between $NH_4^+$ and La Niña.

Revised manuscript:

(1) In Fig. 7B, the signal of $\delta^{18}O$-$NO_3^-$ and $\delta^{15}N$-$NO_3^-$ was plotted to identify the potential sources of $NO_3^-$. Apart from errors introduced from biogeochemical reactions on the signal, according to the isotope composition in different sources, the decomposition of the terrestrial organic matter and its subsequent leaching from soils was an important source of $NO_3^-$ in river waters.

(2) Moreover, because of the overlap in isotope signals (Fig. 7B), fertilizers may also be deemed to be responsible for the DIN yield. In particular, DIN concentrations in Rajang exceeded those detected in pristine tropical black water rivers (Baum and Rixen, 2014), suggesting a potential influence from anthropogenic activities.

(3) Apart from dilution, biogeochemical production for $NH_4^+$ also influences its concentration

(4) Generally, ammonification in soils is strongly dependent on the moisture content (Hopmans et al., 1980). A strong El Niño event was observed from January to June 2016, the Niño 3.4 Index reached 2.5 (threshold 0.5). Subsequently, La Niña occurred and introduced stronger precipitation in Malaysia (Fig. 7D). Consequently, the weather in September 2017 was comparably 'wetter' than the dry season in August 2016. Reichman et al. (1963) observed that $NH_4^+$ concentration in the tested soils decreased after 10-day incubation at high moisture content. Hopmans et al. (1980) found that $NH_4^+$ concentration in the culture soils at the 20% moisture content was markedly higher than the levels from 30% and 35% moisture content groups. Abera et al. (2012) also revealed a significant reduction in extractable $NH_4^+$ content in tropical soils when precipitation enhanced. They addressed that the possible reason for the decreases in ammonification intensity was the enhanced moisture in terrestrial soils, because high moisture could significantly restrain the aeration in peatland and tropical soils and further lead to a depletion of oxygen (Daniels et al., 2012).

Page 9. Section 4.2: First paragraph How much pore water exchange is necessary to explain the observed increase? Is this realistic or are other processes be involved?

Reply: Thank you. The magnitude of pore water exchange is difficult to estimate. Generally, there are many driving forces for this process (Santos et al., 2012; The driving forces of porewater and groundwater flow in permeable coastal sediments: A review; Estuarine, Coastal and Shelf Science). In the estuarine, tidal pumping, wave actions, density difference and bioirrigation could be the driving forces. The rate is highly variable. For instance, the exchange rate caused by bioirrigation could reach 9.5 $m^3$ $m^{-2}$ $d^{-1}$ in Danish sublittoral sediments (Santos et al. 2012), indicating an injection of 1150 mmol $NH_4^+$ $m^{-2}$ $d^{-1}$ in the Rajang estuary (pore water from peat; Table 1). The water depth in the Rajang channels ranged from 5 to 10 meters. Consequently, the $NH_4^+$ addition rate caused by pore water exchange can be 115 to 230 mM $d^{-1}$. The water residence time in the estuary was less than 10 days (unpublished data, from Prof. Moritz Müller and Dr. Aazani Mujahid's research group, calculated by Mohd Fakharuddin Muhamad; all are co-authors of this manuscript). Adding these together, the $NH_4^+$ concentration elevation at this circumstance can be 11.5 to 23 mM, which is significantly higher (approximately three orders of magnitude higher) than the value obtained the cruise. In the Rajang area, the worms and crabs were frequently observed, with a high density. As a result, the exchange rate should not be very low. Moreover, pore water exchange could cover a large

area (Santos et al., 2012), indicating a concentration increase on the regional scale, which fits the pattern gained in the cruises.

However, there might be several other processes involved in the mixing. In the revised manuscript, we modified our expressions, highlight the possibility, not the certainty. In addition, we showed the presence of two reasons at the beginning of Part 4.2.

Revised manuscript: Coupled with the macro-meso tides in Rajang estuary, the magnitude of $NH_4^+$ flux from pore water to the Rajang might be great, as outlined in Fig. 8A, likely to be the main reason for the concentration elevation.

This concentration increase generally result from (1) a direct input from tributary streams or pore water exchange and (2) N transformations.

Page 9. Section 4.2: second paragraph You claim that PN is not involved in the transformation processes, but given the high PN concentrations and low DIN concentrations, small changes in PN may have a large impact on DIN. I suggest to do some simple calculations, how much PN has to be reduced to explain the observed DIN changes. See also comment to page 10, line 33

Reply: Thank you. Indeed, compared to the DIN concentration, the PN levels are higher. We could make a calculation on the basis of Aug 2016 because of the active transformation. As displayed in Fig. 7E, the offset of $NO_3^-$ in Aug 2016 was ca. 10 μM, while $NH_4^+$ offset frequently ranged from 2 to 4 μM. Adding these together, the net increase for DIN is likely to be 10 to 14 μM in Aug 2016, indicating that more than 50% PN transformation (PN concentration in Fig. S5) if PN is the sole support. This high transformation could be observed on the signal of PN. However, the isotope fraction of PN was similar among each site, suggesting a weak reaction. Moreover, for the more sensitive parameter, such as the Left/Right style of Amino acids of the suspended particles, we did not observe significant variations compared to the conservative mixing (Zhu et al., plan to submit to the same issue). The low reactivity of suspended particles in the estuary may be due to the occurrence of degradation in the upper stream. In the revised manuscript, we mentioned this reason in the context.

Revised manuscript: In the mixing experiment, small differences in DON concentration between groups were found, indicating the decomposition of PN was weak. The oxic consumption of these particles in the upper stream might be the reason for the low reactivity for particles in the degradation potential. Therefore, the presence of high concentrations of PN cannot benefit $NO_3^-$ addition.

Page 9, line 32/33. This sentence reads as if DON in the mixing zone is lower than in the coastal ocean. Please rephrase.

Reply: Thank you. We have revised the sentence into 'In the Rajang estuary, despite the injection from sediment pore water (Fig. 8A), net DON consumptions were obtained.'

Page 10. Line 10:*: : :* their input can be identified. This is just a claim. Please substantiate.

Reply: Thank you. We believe this comment belongs to Page 9, line 10. In the revised draft, we added more references and demonstrations to support the influence of these streams to Rajang is limited.

Revised manuscript: Compared with Rajang, these streams were not identified as the major surface flows in Sarawak (Sa'adi et al., 2017), due to the small discharge rates. The influence

from these small loadings can be rapidly diluted in the estuary based on the mathematical simulation (Allen, 1982). In addition, the carbon content ($p$CO$_2$) and dissolved organic carbon did not show significant peaks in the outlet of these stream during the survey (Müller-Dum et al., 2018; Martin et al., 2018). Consequently, the influence introduced by these streams might be minor importance.

Page 10, line 20 – 28. This part of the discussion could fit better in Part 4.1
Reply: Thank you. In Part 4.1, we mainly focus on the DIN sources in the fresh river water (salinity 0). In Part 4.2, we discuss transformations in Rajang branches. In line 20 to 28, we discussed the deforestation and difference between peat and non-peat areas in the estuary. Consequently, we put this section in Part 4.2. According to this suggestion, we believe we the focus of Part 4.1 might not be very clear. In the revised draft, we added a sentence at the beginning of Part 4.1, i.e. 'In the present study, we collected samples in the drainage basin (river water, salinity 0) and the estuarine.' It defines the focus of 4.1 (River water, salinity 0). In addition, in the revised draft, we changed the title of 4.1 into 'DIN sources in Rajang river water', which also highlights the focus.

Page 10, line 33 Here you claim that PN can play a role, but in 4.1 you claim that PN does not play a role. Please clarify this.
Reply: Thank you very much. In Part 4.1, we discussed mineralization of PN in the upper stream water, indicating the potential effect of PN in the DIN circling. Because the water we chose is fresh river water (salinity: 0), the contribution of PN may be limited in the upper zone. In the estuary, the degradation of PN might be weak due to reactions in the upper stream. In Page 10, line 33, we highlighted two possibilities that PN could be involved in the DIN circling. In the following passage, we explained that the PN-derived addition might be weak, while the denitrification on the particle surface might be strong. In the revised draft, we slightly modified the manuscript, highlighting that PN addition and removal are two possibilities.
Revised manuscript: The second factor that influences the generation of NO$_3^-$ and NO$_2^-$ in the estuary water might be SPM related biogeochemical reactions because suspended particles are versatile. On the one hand, the N content on the particle could release into the water via decomposition (Brandes et al., 2007), subsequently increasing DIN concentration. On the other hand, the suspended particles could provide a large number of anoxic micro-niches for denitrifiers coupled with the oxic degradation (Jia et al., 2016). Together with sediment denitrification, the NO$_3^-$ removal in the estuary could occur. Consequently, the addition or removal (two possibilities) for NO$_3^-$ content in estuary water likely depends on the reaction capability of these two controversial pathways.

Page 11. Line 6 No supporting parameters like chlorophyll are presented that may clarify changes in PN quality. In this respect, I wonder about whether phytoplankton blooms occur? After all, fresh readily degradable organic matter is needed to create the anoxic microniches needed for denitrification. Please clarify this. Also, can you discern between sediment denitrification and water column denitrification?
Reply: Thank you. The Rajang water is very cloudy. The following photo is captured during the cruise. The water is very turbidity, especially in wet seasons.

[Figure]

Consequently, the terrestrial particles are the dominant composition in the entire estuary. The algae blooms were not observed. In the present study, concentration of chlorophyll was not included. In the same cruise, Dr. Patrick Martin measured the concentration and the data can be found in Martin et al., 2018 (same special issue). From their results, we can see that the concentration was low ($<3$ μg $L^{-1}$). In addition, PhD student, Edwin Sia, in Prof. Moritz Müller research group, measured the algae species during the cruises. The biomass in the estuary was much smaller than that in the coastal water. In the revised draft, we added the information that the PN was mainly from upper stream. The chlorophyll concentration was very low, on the basis of Martin et al. (2018).

For the formation of anoxic microniches, indeed, it requires the oxidation on the surface. In the revised draft, we included this information.

For the sediment denitrification and water column denitrification, generally, the sediment denitrification leads to decreases in $NO_3^-$ concentration, while the isotope fractions may not be changed due to the completed removal (Yan et al., 2017; detailed information in the reference list). The water column denitrification could decrease $NO_3^-$ concentration, but increase isotope fraction. In the natural environment, both processes exist. It is very difficult to identify the proportion. In the revised draft, we highlighted the possibility of co-existence of these pathways. Revised manuscript: (1) In the Rajang estuary, the PN percentage in SPM frequently ranged from 0.1% to 0.3%, mainly terrestrial-derived solids because of the low concentration of chlorophyll (Martin et al., 2018), smaller than other tropical rivers located in adjacent regions, e.g. the Wonokromo River (0.5%) and the Rorong River (0.85%) at Indonesia (Jennerjahn et al., 2004), the Godavari River (0.36%; Gupta et al., 1997)...

(2) On the other hand, the suspended particles could provide a large number of anoxic microniches for denitrifiers coupled with the oxic degradation (Jia et al., 2016). Together with sediment denitrification, the $NO_3^-$ removal in the estuary could occur. Consequently, the addition or removal for $NO_3^-$ content in estuary water likely depends on the reaction capability of these two controversial pathways.

Page 25, line 25. This observation reinforced*: : :: : :: : ::* please add a citation to back this statement.

Reply: Thank you. We believe this comment belongs to Page 11. In the revised manuscript, we added two references (Daniels et al., 2012; Xu et al., 2013) to support this statement. The detailed information of these two references was provided in the reference list.

Revised manuscript: This observation reinforced that the biogeochemical reactions in the tropical zone are mainly constrained by precipitations (Daniels et al., 2012; Xu et al., 2013) and hence the global climate events markedly influence the N transformations on a local scale.

Section 4.3 Formula (4) should be transferred to the Material and methods section. Of course, the total loads are strongly dependent on discharge. For that reason I suggest no to focus on loads but on the concentrations: What are the factors responsible for the observed rather low DIN loads???

Reply: Thank you. In the revised draft, we moved the formula 4 into the Material and methods section. In addition, we also changed the order of the figures in the supplemental materials. Indeed, the loading of DIN is dependent on the discharge rate. In this section, we tried to offer a gross estimation on the loading, which may benefit the budget calculation for the South China Sea or global scale. Compared with the reactions in the estuary, this section should not be the key. Consequently, we tried to reduce the length of this section according to your suggestion. For the potential effect, currently, the harmful algae blooms were barely observed. These likely result from the low concentration of DIN and mild input. However, the anthropogenic influences in the Rajang watershed are increasing. The effect may be enhanced in the future. Considering this part is the deduction for the future, in the revised manuscript, we changed the title of Part 4.3 to 'DIN fluxes and implications'.

---

## Referee Report (RR1)

I congratulate the authors with this revised version. In my view all points raised were adequately dealt with.

I still found some language issues in the ms, shown below. And I do suggest the authors to critically look for misspellings and unclear sentences.
Yours,..

P3L20. Over ->above

P3L22 composed by -> of

P5L6 Grassholf -> Grasshoff

P6L3-4 For the magnitude of DIN fluxes (Q) that transported to the coastal ocean, it can be estimated according to the following equation: Replace by
TDIN fluxes (Q) transported to the coastal ocean were estimated according to the following
equation:

P6L25:unsaturated ->undersaturated

P11L28: ....found, indicating (insert that)

P11L29: benefit -> invoke??

P11L31: accelerate (delete s)

P12L12: ...worth noticing that (replace that with the)

P12L20 replace productivity with producers

P12L31. phytoplankton species: replace species with group

P12L32replace "is flourished" with "flourishes"

---

## Author Response (AR2)

Responses to Reviewers

1st Reviewer

I congratulate the authors with this revised version. In my view all points raised were adequately dealt with. I still found some language issues in the ms, shown below. And I do suggest the authors to critically look for misspellings and unclear sentences.

Reply: Thank you very much. In the revised draft. We addressed all these questions from the feedback and made corrections accordingly. In addition, we also carefully went over the draft again and made some additional changes.

1.  P3L20. Over ->above

Reply: Thank you. We have corrected this.

2.  P3L22 composed by -> of

Reply: Thank you. We have changed this.

3.  P5L6 Grassholf -> Grasshoff

Reply: Thank you. We have changed it.

4.  P6L3-4 For the magnitude of DIN fluxes (Q) that transported to the coastal ocean, it can be estimated according to the following equation: Replace by TDIN fluxes (Q) transported to the coastal ocean were estimated according to the following equation:

Reply: Thank you. In the revised draft, we have replaced the sentence accordingly.

5.  P6L25: unsaturated ->undersaturated

Reply: Thank you. We have changed the expression in the revised manuscript.

6.  P11L28: ....found, indicating (insert that)

Reply: Thank you. We have added that in the sentence.

7. P11L29: benefit -> invoke??

Reply: Thank you. We would like express that PN could not increase nitrate addition. In the revised draft, we changed benefit to enhance.

8. P11L31: accelerate (delete s)

Reply: Thank you. We have removed s from this word.

9. P12L12: ...worth noticing that (replace that with the)

Reply: Thank you. We have changed 'that' to 'the' in the revised draft.

10. P12L20 replace productivity with producers

Reply: Thank you. We have changed it (P12, L27 now).

11. P12L31. phytoplankton species: replace species with group

Reply: Thank you. We have changed 'species' to 'group'.

12. P12L32replace "is flourished" with "flourishes"

Reply: Thank you. We have changed it.

2$^{nd}$ Reviewer

I am happy to find that authors made corrections according to our suggestions. I do not have any major questions about the manuscript. Before publication, I believe authors could improve the draft with some minor changes.

Reply: Thank you. We modified the manuscript according to these suggestions.

For main text:

1. Line 12, Page 2: please add δ in front of 15N-PN

Reply: Thank you. We have changed it.

2. Line 11, Page 3, please add 'the' in front of El Niño-Southern Oscillation

Reply: Thank you. We have changed it.

3. Line 11, Page 8: It should be 'Isotope fractions were'.

Reply: Thank you. We have changed it.

4. Line 29, Page 9, Both Müller-Dum et al., 2018; Martin et al., 2018 have published, please add volume and page number in the reference list.

Reply: Thank you. We added the information in the reference list and made changes in the main text.

5. Line 21, Page 10: Passive voice should be better, 'DON may be continuously transformed to…'

Reply: Thank you. We have used passive voice in the revised draft.

For figures

1. Figure 1 legend, please change 'in the Borneo' to 'in Borneo'

Reply: Thank you. We have changed this expression.

2. Figure 2 legend, please remove 'isotope fraction'

Reply: Thank you. We have removed it.

3. The authors used 'the Rajang estuary' in the manuscript, in some legends, they used 'the Rajang Estuary' instead, please change them.

Reply: Thank you. We used 'the Rajang estuary' in all legends in the revised draft.

For Table

1. I did not find Lida et al. 2007 in the reference list, I found Iida et al. 2007 who took a study in the Mekong River, please check the reference.

Reply: Thank you. We have changed it.

For Supply:

1. Authors added a photo in the response to describe the cloudy water in the Rajang River. I believe they could include the photo(s) in the supply. The readers will gain a clearer vision on the Rajang River.

Reply: Thank you. In the revised draft, we added two photos in the supply (Fig. S1) and invoked these photos in the main text.

2. Table S1, please use italic 'f'

Reply: Thank you. We have modified it.

3. Table S1, I believe the riverine $NO_3^-$ concentration is Fig. 7, not Fig. S9 to S11

Reply: Thank you. We have changed it.